# Brainstem development requires galactosylceramidase and is critical for pathogenesis in a model of Krabbe disease

Nadav I. Weinstock[1,2], Conlan Kreher [1,2], Jacob Favret[1,2,3], Duc Nguyen[4], Ernesto R. Bongarzone[4], Lawrence Wrabetz[1,2,5,6], M. Laura Feltri [1,2,5,6] & Daesung Shin [1,2,3,6 ✉]

Krabbe disease (KD) is caused by a deficiency of galactosylceramidase (GALC), which induces demyelination and neurodegeneration due to accumulation of cytotoxic psychosine. Hematopoietic stem cell transplantation (HSCT) improves clinical outcomes in KD patients only if delivered pre-symptomatically. Here, we hypothesize that the restricted temporal efficacy of HSCT reflects a requirement for GALC in early brain development. Using a novel *Galc* floxed allele, we induce ubiquitous GALC ablation (*Galc*-iKO) at various postnatal timepoints and identify a critical period of vulnerability to GALC ablation between P4-6 in mice. Early *Galc*-iKO induction causes a worse KD phenotype, higher psychosine levels in the rodent brainstem and spinal cord, and a significantly shorter life-span of the mice. Intriguingly, GALC expression peaks during this critical developmental period in mice. Further analysis of this mouse model reveals a cell autonomous role for GALC in the development and maturation of immature T-box-brain-1 positive brainstem neurons. These data identify a perinatal developmental period, in which neuronal GALC expression influences brainstem development that is critical for KD pathogenesis.

[1] Hunter James Kelly Research Institute, Jacobs School of Medicine and Biomedical Sciences, University at Buffalo (SUNY), Buffalo, NY 14214, USA. [2] Department of Biochemistry, Jacobs School of Medicine and Biomedical Sciences, University at Buffalo (SUNY), Buffalo, NY 14214, USA. [3] Department of Biotechnical and Clinical Laboratory Sciences, Jacobs School of Medicine and Biomedical Sciences, University at Buffalo (SUNY), Buffalo, NY 14214, USA. [4] Department of Anatomy and Cell Biology, College of Medicine, University of Illinois at Chicago, Chicago, IL 60612, USA. [5] Department of Neurology, Jacobs School of Medicine and Biomedical Sciences, University at Buffalo (SUNY), Buffalo, NY 14214, USA. [6] Neuroscience Program, Jacobs School of Medicine and Biomedical Sciences, University at Buffalo (SUNY), Buffalo, NY 14214, USA. ✉email: daesungs@buffalo.edu

Krabbe disease (KD) is a demyelinating and neurodegenerative lysosomal storage disorder (LSD), often fatal in infancy[1]. KD is caused by mutations in GALC, a galactolipid hydrolase that must be trafficked to the lysosome for proper functioning[2]. GALC is involved in the normal turnover of myelin by hydrolyzing galactosylceramide, a major sphingolipid constituent of myelin which is important in myelin compaction[3]. Unlike other LSDs, the primary substrate of GALC, galactosylceramide, does not accumulate broadly in KD tissues. Instead, a minor substrate of GALC, psychosine (galactosylsphingosine), accumulates to toxic levels, causing extensive demyelination and the bulk of KD pathological findings (psychosine hypothesis[4]). In support of this hypothesis, in vitro studies show that treating cells with psychosine increases proapoptotic factors and kills oligodendrocytes (OL)[5], Schwann cells[6,7], and neurons[8]. The long-held assumptions of the psychosine hypothesis were recently confirmed, in vivo, demonstrating that the abnormal accumulation of psychosine is toxic and is generated catabolically through the deacylation of galactosylceramide by acid ceramidase[9].

More than 85% of KD patients exhibit the rapidly progressive infantile-onset form of disease, which leads to death by 2 years of age. Although there is no cure for KD, hematopoietic stem cell therapy (HSCT) attenuates neurologic deterioration and improves developmental gains[10]. These benefits are particularly sensitive to the severity of disease at transplantation, and are only beneficial if delivered at a clinically defined presymptomatic timepoint[10]. Intriguingly, pre-clinical gene therapy trials in twitcher mice are also time-sensitive, and have been shown to be more efficacious if delivered shortly after birth[11]. These data therefore suggest a presymptomatic therapeutic window, in which treatment of KD with HSCT or gene therapy is more efficacious. Why treatment must occur so early remains unknown. While GALC should intuitively be required during myelination, many of these temporal events seem to precede the bulk of CNS myelination. Furthermore, patients diagnosed in utero, and treated within the first few weeks of life, do better than patients treated at 1–2 months old[12,13]. These findings underlie the necessity for properly defining the precise timepoint in which GALC is first required.

In a previous study, we used a global metabolomic analysis of twitcher hindbrains to detect the earliest biochemical changes that occur in KD pathogenesis[14]. While we saw many metabolic changes at an early symptomatic stage of disease (P22), we were surprised to find a number of biochemical processes that were also significantly altered at a presymptomatic timepoint (P15). These changes, though subtle, reflected diverse neuro-metabolic functions relating to glycolysis, the pentose phosphate pathway, hypoxanthine metabolism, and mannose-6-phosphate, an important residue involved in lysosomal enzyme trafficking[14]. These findings, along with the observed presymptomatic therapeutic window, led us to hypothesize that GALC has important and specific functions in the early brain development.

Several ubiquitous Galc mutant mice have been used to study KD, including twitcher[15], twi-5J[16], humanized GALC transgenic[17], GALC-Gly270Asp[18], Galc-His168Cys knock-in[19], and Saposin A knockout (KO) mice[20]. Although these models were instrumental in the characterization of KD and the development of various therapeutic strategies, none of them could identify the temporal effect of GALC deficiency on the progression of KD. We therefore engineered a conditional Galc floxed mouse by gene targeting, which provided us with the opportunity to directly ask at which age is Galc required. Both constitutive Galc-KOs and induced Galc-KOs (iKO) generated from the conditional floxed allele recapitulated a range of neurologic features seen in KD patients. Our study revealed a key developmental process that requires GALC in the perinatal period. Induced deletion of Galc prior to P4 resulted in severe neurodevelopmental defects that were particularly profound in the brainstem. Conversely, deletion of Galc after P6 resulted in prolonged survival and attenuated pathology. This study demonstrates that temporal GALC expression is likely a major contributor to brainstem development. Augmenting GALC levels at, or prior to, this newly defined perinatal period would likely improve the efficacy of therapeutic interventions for KD.

## Results

**The Galc knockout mouse is an authentic model of KD**. To understand the temporal requirements of GALC, and its relationship to the progression of KD, we developed a conditional Galc floxed mouse (Supplementary Fig. 1a), maintained on a congenic C57BL/6 background. To determine if these mice could be used to accurately model KD, global Galc-KO mice derived from the Galc floxed allele were directly compared to the well-studied twitcher (Galc^W339X) mice[15]. Galc-KO mice were generated by crossing Galc floxed mice to CMV-Cre, in which Cre is expressed ubiquitously[21]. PCR analysis of total brain genomic DNA revealed that the Galc gene was efficiently deleted (Fig. 1a). Northern blot analysis of total brain RNA from twitcher (Galc^twi/twi), Galc-KO (Galc^−/−), and wild-type (WT) littermates (Galc^+/+) showed that Galc transcripts were entirely removed from the Galc-KO brain (Fig. 1b). Like twitcher, homozygous Galc-KO had no GALC activity (Fig. 1c) and developed the same phenotype as twitcher[22,23], namely severe motor coordination defects, a reduced life span of ~45 days and attenuated growth beginning at P21 (Fig. 1d–g). Both Galc-KO and twitcher mice had an increase in markers of brain inflammation, such as toll-like receptor 2 (TLR2), CD68/CD163, Iba1 (all detect microgliosis), and glial fibrillary acidic protein (GFAP; astrogliosis; Fig. 1h, i). Taken together, the novel Galc-KO model has a KD-like phenotype similar to twitcher and is a true GALC-null allele.

**Survival of CAG-Cre/ER^T driven Galc-iKO mice is dependent on the timing of Galc deletion**. To determine the age by which Galc must be deleted to trigger KD pathogenesis, we crossed Galc floxed mice with inducible ubiquitous Cre mice (CAG-Cre/ER^T)[24] to generate Galc-iKO (Fig. 2a). To maximize the efficiency of Galc ablation, we used haplodeficient Galc^flox/null(−) mice. Galc-iKO mice were induced between P0 and P10. As expected, perinatal induction of Galc ablation (occurring on the day of birth, postnatal day 0) produced significant neurologic impairments that closely mirrored the Galc-KO. These mice developed robust neurologic symptoms by P35, including irritability, tremors, wasting, hindlimb paralysis, and ultimately death by P60 (Fig. 2b, c). Galc-iKO mice induced between P1 and P4 (hereafter Galc-iKO ≤ P4) exhibited a similar clinical phenotype and also survived to 60 days. Surprisingly, when Galc deletion was initiated between P6 and P10 (hereafter Galc-iKO ≥ P6), the mice developed a significantly protracted clinical course and survived until P90 (Fig. 2b, c). Galc-iKO ≥ P6 had no obvious neurological phenotype or somatic growth defects until around P60 (Fig. 2c; iKO P8). Compared to Galc-iKO ≤ P4, Galc-iKO ≥ P6 had significantly less KD pathology at P60, including fewer inflammatory globoid cells (Fig. 2d–f). These data suggest that Galc ablation at an early developmental timepoint dramatically influences the clinical course of KD and overall degree of pathology produced.

**The CAG-Cre/ER^T transgene recombines efficiently with tamoxifen, regardless of induction time**. We were concerned that the differential survival of Galc-iKO mice could be due to technical limitations of our inducible system caused by temporal variability in recombination efficiencies. To quantitatively determine the temporal recombination efficiency of our inducible

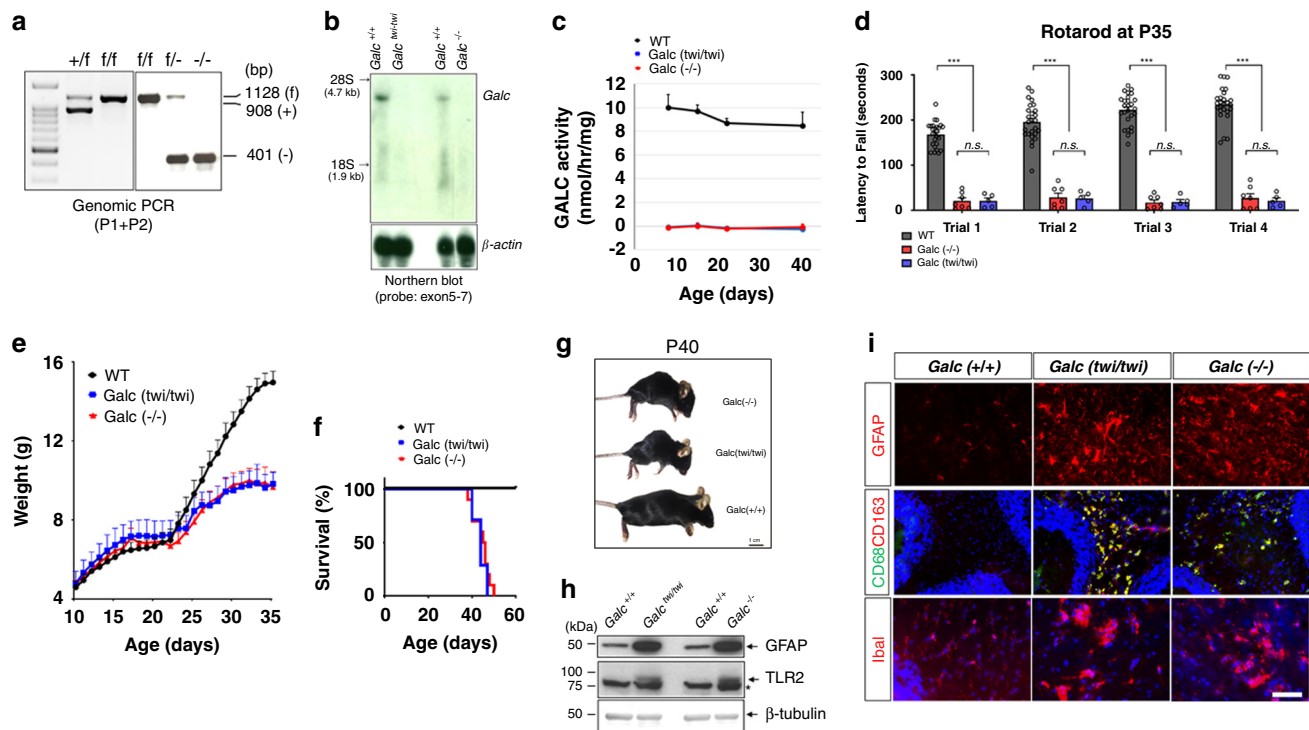

**Fig. 1 The *Galc*-KO (*Galc*⁻/⁻) is an authentic murine model of KD, analogous to *twitcher* (*Galc*^twi/twi^).** **a** PCR-based genotyping of mice (wild-type (WT) (+), conditional (f), and KO (−)) using the P1–P2 primer pair in Supplementary Fig. 1a. *Galc*-KO was generated by mating the *Galc* conditional allele (f) to a ubiquitous Cre (CMV-Cre). The floxed locus is efficiently recombined. **b** Northern blot analysis of total brain RNAs from P30 *twitcher* (*Galc*^twi/twi^), *Galc*-KO (*Galc*⁻/⁻), and WT littermate (*Galc*⁺/⁺). The probe spanning exons 5–7 of *Galc* cDNA showed that *Galc* transcripts were successfully removed from the *Galc*-KO, similar to the situation in the *twitcher*. The experiment was repeated twice with samples from different animals. **c** There were no remaining GALC activities in the brains of both the *twitcher* and homozygous *Galc*-KO; n = 4. Data are presented as mean values ± SEM. **d** Rotarod analysis of P35 animals showed that both *twitcher* and *Galc*-KO had the same poor performance. n = 25 for WT, 7 for KO, and 5 for *twitcher*. One-way ANOVA with Bonferroni post hoc multiple comparison test was used. Data are presented as mean values ± SEM; ***p = 0.000001. **e** Both mice showed a reduced growth curve after P21. n = 10 each genotype (**f**) *Galc*⁻/⁻ survived ~45 days, like *Galc*^twi/twi^; n = 10. Data are presented as mean values ± SD; *p < 0.05, **p < 0.01, and ***p < 0.001; ns not significant. **g** Body size of moribund *Galc*-KO and *twitcher* mice was much smaller than WT. **h** Western blot analysis revealed dramatic increases in the markers of astrocytosis (glial fibrillary acidic protein; GFAP) and activated microglia (toll-like receptor 2; TLR2) in the brain of *Galc*⁻/⁻ as *Galc*^twi/twi^ compared to WT *Galc*⁺/⁺. Asterisk (*) is a nonspecific band. **i** Immunohistochemistry on cryosections of cerebellum white matter showed activated astrogliosis (GFAP) and activated microglia (Ibal, CD68, and CD163), in the brains of both *Galc*⁻/⁻ and *Galc*^twi/twi^. Scale bar = 100 μm. DAPI is blue colored. The experiment was repeated three times with samples from different animals. Animals in **g–i** were P40.

system, we used a tdTomato Cre reporter[25]. We began by crossing the tdTomato transgene with the CAG-Cre/ER^T^ allele and induced recombination with tamoxifen at P3 (*Galc*-iKO ≤ P4) or P8 (*Galc*-iKO ≥ P6). Both timepoints showed an equal number of tdTomato-positive cells and tdTomato protein levels in brain (Fig. 3a–c), suggesting similar recombination efficiencies. Furthermore, GALC enzymatic activity was similarly absent from the brains of *Galc*-iKO mice induced early and late (Fig. 3d), with only 2% of WT GALC activity remaining. This finding was likewise confirmed by immunofluorescence for GALC expression before and after induction of early and late timepoints (Fig. 3e, f). Similarly, peripheral tissues including sciatic nerve, liver, and spleen had only 6%, 1–4%, and 5–7% of WT GALC activity remaining, respectively (Supplementary Fig. 2a, c). We also considered that GALC protein could be particularly stable, and could theoretically exist well beyond the induction of *Galc* DNA ablation. We therefore determined the half-life of GALC protein, in vitro, and found it to be ~3 h (Fig. 3g), fitting well with the near total reduction in GALC activity seen at both timepoints, in vivo (Fig. 3d). These data collectively suggest that technical issues regarding differences in inducing recombination likely do not explain the altered survival and clinical course surrounding temporal GALC ablation.

Finally, we asked if the reduction in GALC activity persisted equally when induced early or late. While there was no significant difference in the residual GALC activities in the brains of animals induced at different times, we were surprised to find that ~20% of GALC activity eventually returned in all late-stage, moribund *Galc*-iKO brains (Fig. 3h). A similar increase of GALC activity was also observed in the sciatic nerve (12–14%) and marginally in the liver (5–9%), but not at all in the spleen (3–7%; Supplementary Fig. 2b, d). Cre/ER^T^ is only active as long as tamoxifen persists[26]; suggesting that a small portion of residual WT cells escaped Cre recombination and were able to proliferate and restore some levels of GALC activity. This return of GALC activity is particularly intriguing as these levels were surprisingly not sufficient to prevent or rescue KD pathology. We theorize that the lack of rescue may be explained by GALC levels that are too low, expressed in the wrong cell type, or present at the wrong time period. Furthermore, cross-correction of GALC[27] (and other lysosomal hydrolases[28]) may not be very efficient in vivo, and therefore, the function of GALC may be restricted to the subset of cells that directly express it. A similar correlate exists regarding refractory moribund pathology observed in HSCT-treated *twitcher* and KD patients despite the detection of substantial GALC activity[29–32]. Together, these observations further

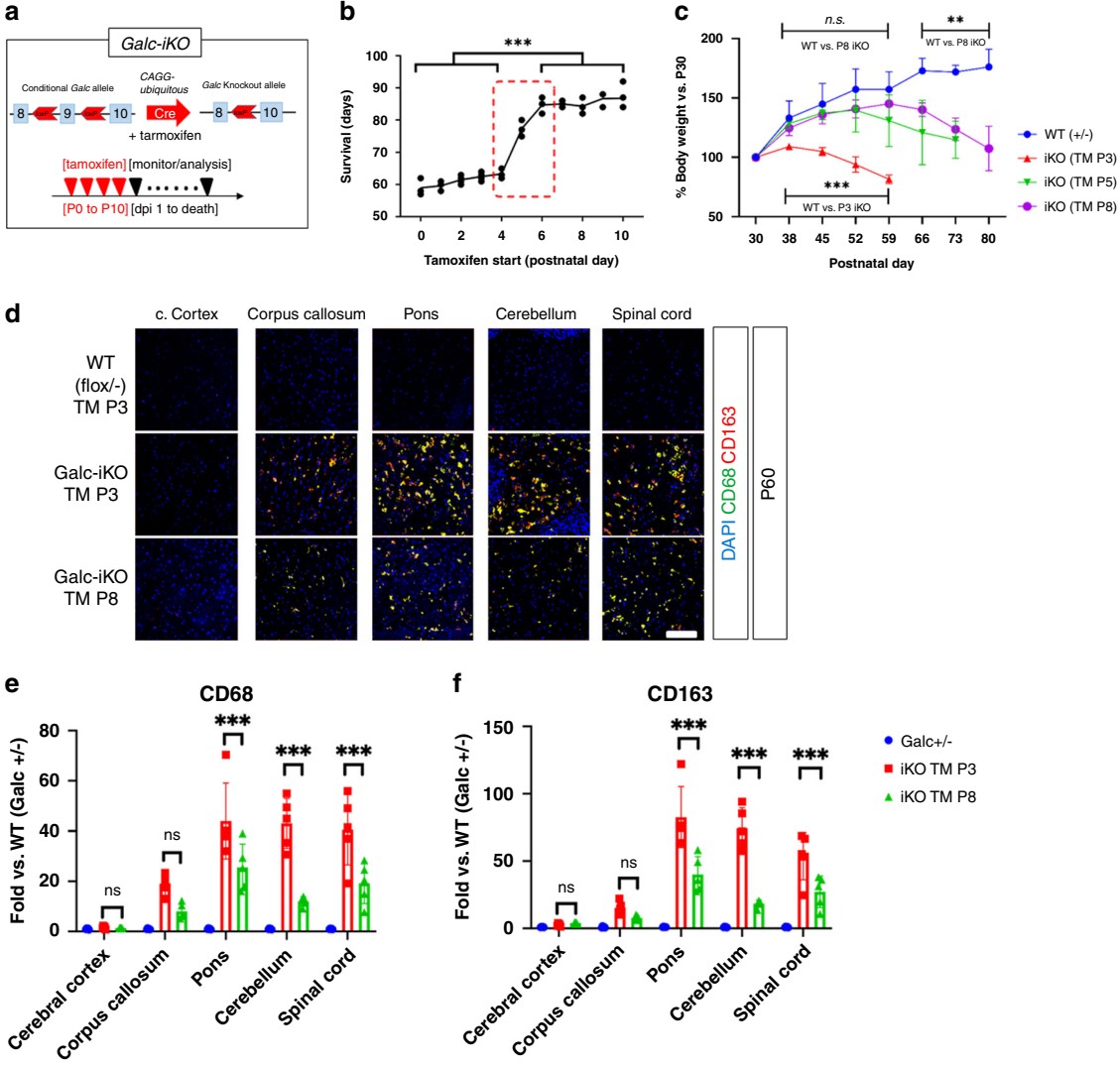

**Fig. 2 Differential survival of CAG-Cre/ER^T driven *Galc*-iKO mice depending on the starting time of tamoxifen injection. a** *Galc* floxed mice were crossed with inducible ubiquitous Cre mice (CAG-Cre/ER^T), and induced with tamoxifen for four consecutive days. **b** When the deletion started at or before P4, the mice only survived ~60 days, but when *Galc* deletion was induced at or after P6, the mice had no phenotype until P60, started to show a tremor after P60 and survived almost 3 months; $n = 6$ per age. One-way ANOVA test was used; ***$p = 0.000001$. **c** Body weight of *Galc*-iKO mice were drastically reduced at or before 20–25 days from the last injection; $n = 6$ per genotype. Data are presented as mean values ± SD. One-way ANOVA test was used. **d** The late-induced mice (TM starting at P8) were not as severe as the early-induced *Galc*-iKO (TM at P3), when the latter were moribund at P60. The experiment was repeated three times with samples from different animals. Inflammation markers CD68/CD163 levels were significantly lower in the late-induced brains compared to those of the early-induced iKO (**e**, **f**); $n = 6$ per genotype. Two-tailed unpaired Student's *t* test was used. Data are presented as mean values ± SD (**b**, **c**) and SEM (**e**, **f**); *$p < 0.05$, **$p < 0.01$, and ***$p < 0.001$, ns not significant. *P* values for CD68/CD163 in the comparisons of *Galc*-iKO P3 vs P8 are 0.9991/0.9705 (cerebral cortex), 0.0508/0.4415 (corpus callosum), 0.0002/0.0001 (pons), 0.0001/0.0001 (cerebellum), and 0.0001/0.0002 (spinal cord), respectively. Scale bar = 100 μm.

emphasize the nuance of providing sufficient GALC to specific cell types at an appropriate temporal period.

**Induction timing of *Galc*-iKO affects the differential accumulation of psychosine.** While mice induced after P6 survived longer than those induced earlier, all *Galc*-iKO mice eventually developed a rapidly progressive neurologic decline. Morphometric analysis of optic nerves from early and late-induced moribund *Galc*-iKO mice exhibited similar degrees of demyelination and axonal degeneration (Fig. 4a–c). This was also true for spinal cord tissues, though a more pronounced difference in axonal pathology occurred between early- and late-induced animals (Fig. 4d–f). Globoid cells and markers of neuroinflammation also accumulated in both end-stage *Galc*-iKO ≤ P4 and *Galc*-

iKO ≥ P6 brains (Fig. 5a–d), though to a slightly higher degree in early-induced mice and especially evident in the hindbrain (pons, cerebellum, and spinal cord). Taken together, moribund *Galc*-iKO mice all develop canonical KD pathology that are qualitatively similar to each other, regardless of induction time.

To elucidate the cause of pathology in moribund mice, we measured psychosine levels in the cervical spinal cord of *Galc*-iKO mice. Psychosine is a highly cytotoxic lipid, capable of inducing cell death of myelinating cells and neurons[8,33,34]; furthermore its accumulation is correlated with severity and onset of disease[35]. High-performance liquid chromatography tandem mass spectrometry (LC–MS–MS) showed 300–350 pmol of psychosine accumulated per mg of protein in the moribund *Galc*-iKO ≤ P4. This was far higher than the psychosine

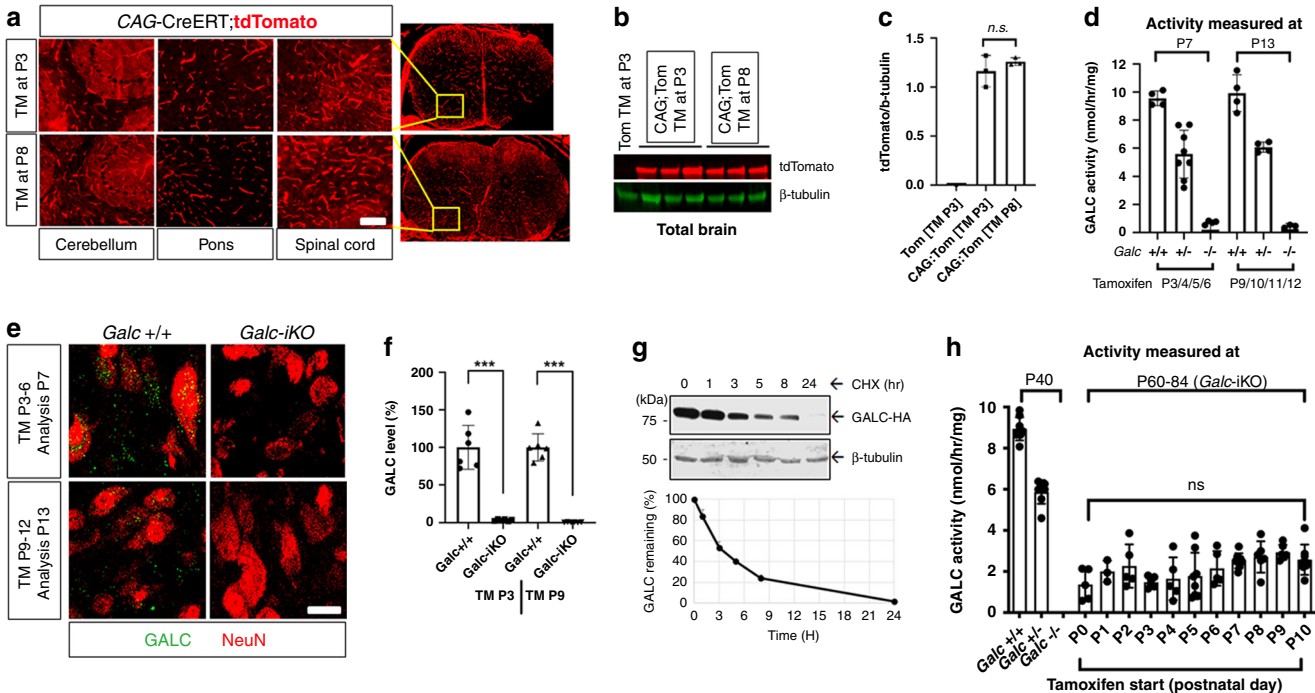

**Fig. 3 The CAG-Cre/ER$^T$ transgene induces recombination efficiently, regardless of induction time. a–c** CAG-Cre/ER$^T$;tdTomato mice were induced with tamoxifen starting at P3 or P8, and dissected at P16. The two induction times have a similar number of tdTomato-positive cells in brain and spinal cord (**a**) and the same level of tdTomato protein in whole brains (**b, c**), suggesting no difference in recombination efficiency. Scale bar = 100 μm; $n = 3$ per genotype. Data are presented as mean values ± SD. **d** Regardless of induction time, brain GALC activity was efficiently removed 24 h post tamoxifen injection (from the last day of four consecutive days), and were only 2% of normal levels in the total brains; $n = 4$ per genotype, except TM-P3 (±) $n = 8$. **e, f** GALC protein was efficiently depleted 24 h after tamoxifen injection in the brains of *Galc*-iKO, regardless of induction timing. The image is from the pons. Scale bar = 20 μm; $n = 6$ per genotype; ***$p = 0.0001$. **g** Western blot analysis of HEK-293T cell lysates transfected with HA-GALC, treated with 100 μg/ml cycloheximide, harvested at the indicated timepoints showed that the half-life of GALC protein, in vitro, is ~3 h. The experiment was repeated three times with samples from different transfections. **h** GALC activity was not detected at the last day of survival in the whole brains of *Galc*-KO mice (P40) when mice were moribund and paralyzed. Surprisingly, when *Galc*-iKO brains were sampled at similar moribund conditions (between P60 and P84), substantial brain GALC activity had somehow returned and was ~20% of normal levels, regardless of induction starting time; $n \geq 3$ per genotype. Data are presented as mean values ± SD (**c, h**) and SEM (**d, f**). Two-tailed unpaired Student's *t* test (**d, f**) and two-way ANOVA with Tukey's multiple comparison tests (**h**) were used; *$p < 0.05$, **$p < 0.01$, and ***$p < 0.001$; ns not significant.

concentration in control spinal cords (5–8 pmol/mg of protein, Fig. 5e). Although psychosine also accumulated highly in *Galc*-iKO ≥ P6 spinal cords (110–150 pmol/mg of protein), these levels were significantly less than the *Galc*-iKO ≤ P4. The level of psychosine in *Galc*-iKO induced at P5 rested between these extremes (150–160 pmol/mg of protein), suggesting that the concentration of psychosine is directly correlated with pathology and survival of *Galc*-iKO mice. Previous studies described that psychosine accumulates in white matter of the CNS, in a caudal-to-rostral pattern, parallel to the progression of myelin development[16,36,37]. To investigate if the timing of GALC depletion affects the region-specific accumulation of psychosine, we further analyzed psychosine levels in subanatomic regions of the CNS, including the cerebral cortex, cerebellum, and brainstem of both end-stage *Galc*-iKO ≤ P4 and *Galc*-iKO ≥ P6. Interestingly, early-induced mice (P3) had higher psychosine accumulation in the brainstem and spinal cord vs the cerebral cortex or cerebellum (Fig. 5f). This region-specific pattern of psychosine accumulation was not present in the late-induced brains (P8). Furthermore, the brainstem and spinal cords of early-induced *Galc*-iKO (P3) had significantly higher psychosine levels than late-induced *Galc*-iKO (P8), though the cerebral cortex and cerebellum did not (Fig. 5f). These data suggest that perinatal GALC expression prevents psychosine accumulation in a brainstem and spinal cord, region-specific fashion.

**GALC expression reaches its highest peak at P5 in the WT brain.** We hypothesized that the differential survival of *Galc*-iKO mice induced at presymptomatic perinatal ages may reflect a role for GALC in early brain development. We began to address this point by seeking to better understand how *Galc* expression is regulated during postnatal brain development. To do so, we analyzed the mRNA transcripts, protein, and activity of GALC in WT brains. Interestingly, in situ hybridization and northern blot analyses showed that *Galc* transcript levels are low at P0, but increase and reach their highest peak at P5 in WT brains (Fig. 6a, b). These peaks correlate closely with the observed clinical worsening and effects on survival that occurred, when GALC was ablated temporally (Fig. 2b). Similarly, GALC enzymatic activity mirrored mRNA patterns and peaked at P5 (Fig. 6c), supporting the idea of a developmental process that requires GALC function. Notably, the expression pattern of *Galc* transcripts by in situ hybridization suggest *Galc* may also be expressed in non-myelin regions of the P5 brain, particularly in neurons (Figs. 6a and 3e). For example, *Galc* transcripts were clearly expressed in P5 hippocampal neurons and granular neurons of the cerebellum (Fig. 6a enlargement). These data are consistent with a number of previous studies that have shown GALC is expressed and is important in neurons[8,38]. The unexpected distribution of GALC encouraged us to carefully determine the cellular expression of GALC in perinatal brain development. We therefore performed GALC colocalization experiments using various cell markers

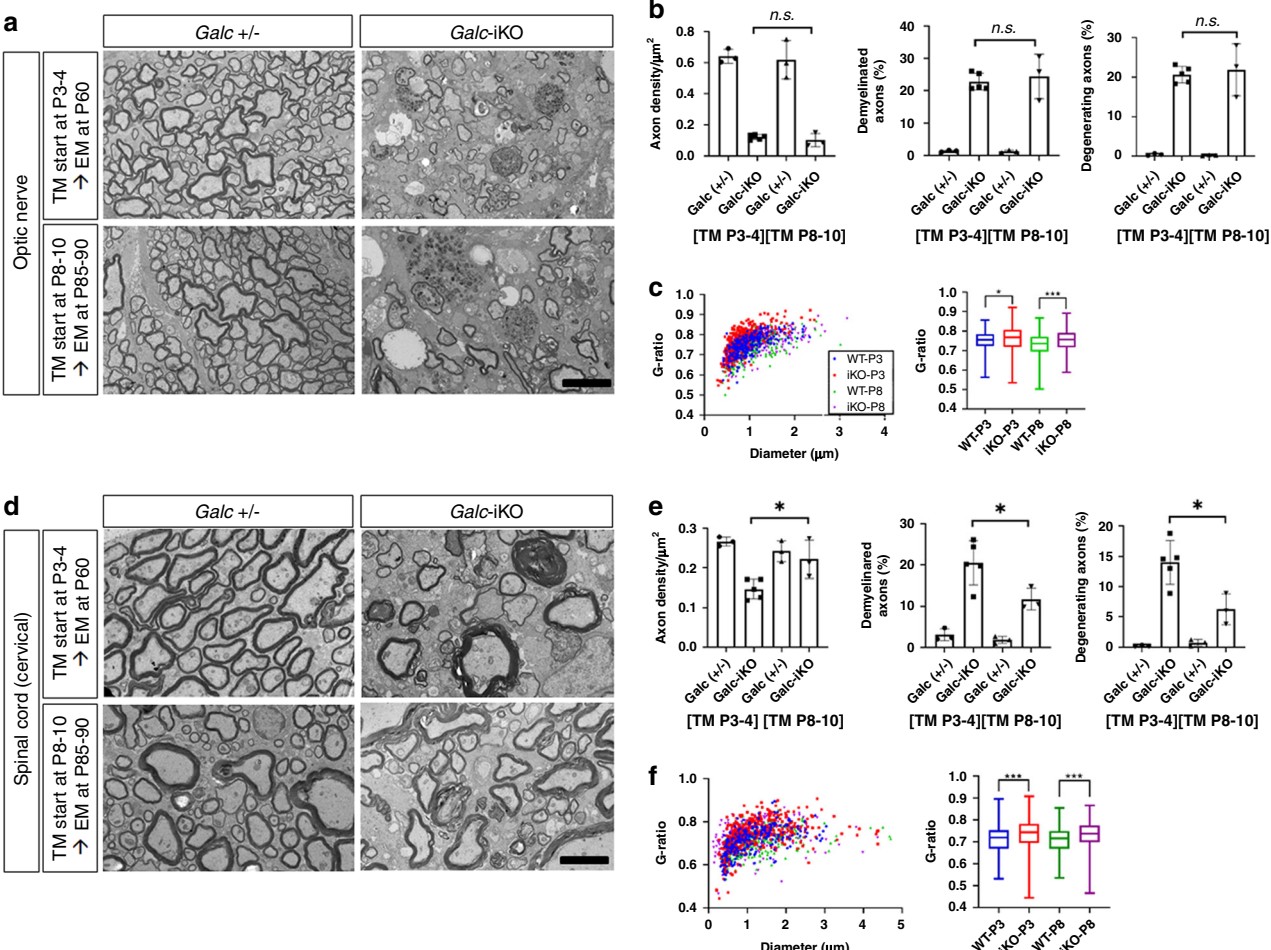

**Fig. 4 *Galc*-iKO brains have morphological signs of demyelination and axonal degeneration regardless of induction starting time.** Electron microscopy analysis of the optic nerves (**a**), spinal cords (**d**), and their morphometry quantifications (**b**, **e**) from *Galc*-iKO and corresponding WT. Induced deletion starting at either P3–4 (*Galc*-iKO ≤ P4) or P8–10 (*Galc*-iKO ≥ P6), showed that both conditional *Galc*-iKO brains have morphological signs of demyelination, axonal degeneration, and gliosis. Although there was no difference of those pathological extents in the optic nerves between the early- and late-induced *Galc*-iKOs, the morphometric pathological parameters, such as degenerating and demyelinating axons, were slightly less present in the spinal cord of *Galc*-iKO ≥ P6 than *Galc*-iKO ≤ P4. Scale bar = 4 μm. Unpaired two-tailed Student's *t* test was used. **c**, **f** The myelin sheaths of both optic nerves and spinal cords were significantly thinner in both *Galc*-iKO ≤ P4 and *Galc*-iKO ≥ P6 vs each WT; *n* = 3, except *Galc*-iKO ≤ P4 *n* = 5. All data are presented as mean values ± SD. Welch's *t* test was used; *$p$ = 0.0237 and ***$p$ = 0.0001; ns not significant. The box bounds in the box plots for the *g*-ratio are 25th and 75th percentiles, the center lines are median value, and the whiskers are 0.05 and 0.95 percentiles.

throughout multiple anatomic regions in the P5 brain. GALC protein was expressed in all brain cell types, including neurons, microglia, astrocytes, and OL-lineage cells (Fig. 6d, e). The majority of GALC at P5 was expressed in neurons (38–73%) and was far higher than the expression in OL-lineage cells (7–11%), astrocytes (5–18%), or microglia (4–7%). The unlabeled GALC-positive population, others (6–44%), presumably reflect undifferentiated cell types[39]. Regional GALC level among cerebral cortex, striatum, pons, and spinal cord was not significantly different at P5, which was assessed by antibody staining (Fig. 6f). Furthermore, neuronal GALC expression was developmentally influenced and peaked at P5 (Fig. 6g, h). These results suggest a role for GALC in early neuronal development.

**Brainstem development requires GALC expression.** The parallel findings of increased GALC expression at P5, along with increased susceptibility to deletion of GALC prior to P5, suggests that a developmental process dependent on GALC occurs at P5. This finding is particulary surprising, as *twitcher* symptomatology does not begin until P21, and therefore suggests the possibility of

an early developmental defect occuring prior to hallmark of KD progression. To monitor the neuronal development, we used Thy1.1-YFP reporter mice that express YFP at high levels in a sparse population of motor, sensory, and some central neurons[40]. Thy1.1-YFP was crossed with *Galc*-KO mice or WT controls. Confocal analysis showed a dramatic decrease in the density of YFP axons in the brainstem of P35 symptomatic *Galc*-KO;Thy1.1-YFP mice (Fig. 7a). Coronal sections confirmed that pyramidal, pontine retucular, trigeminal, and gigantocellular nuclei were dramatically affected in the pons and medulla of *Galc*-KO;Thy1.1-YFP mice (Fig. 7b:1). Instead, neurons of the cerebellum were only moderately changed, while other brain rostral regions were not affected (Fig. 7b:2, 3). High power neuronal magnification further emphasized the region-specific nature of pathology seen in the *Galc*-iKO mice (Fig. 7c, d), ultimately suggesting that hindbrain pathology is likely a major consequence of KD pathogenesis. A protein marker for neuronal processes, Tuj1 (neuron-specific class III beta-tubulin), was also similarly reduced in the brainstem of P35 moribund *Galc*-KO compared to WT (Supplementary Fig. 3a). Mutant axons sometimes appeared

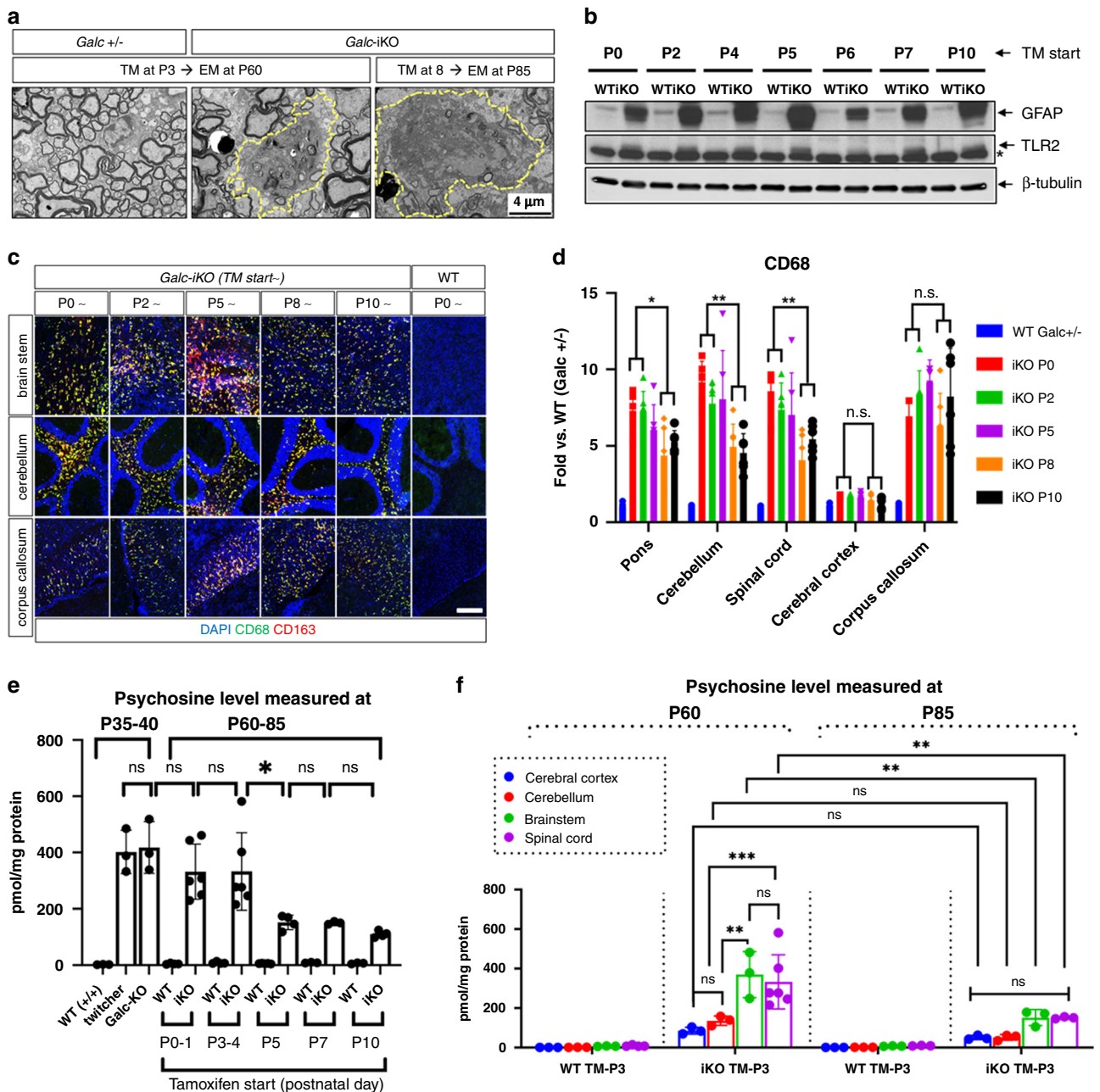

**Fig. 5 Moribund *Galc*-iKO mice exhibit qualitatively similar KD pathology.** Regardless of induction starting time, the moribund *Galc*-iKO mice had the appearance of globoid cells (**a**) and increased of inflammation proteins, such as GFAP and TLR2/CD68/CD163 (**b**, **c**), compared to WT littermate (*Galc*+/+). Asterisk (*) is a nonspecific band. Scale bar = 100 μm. The experiment was repeated three times with samples from different animals. **d** Quantitative analysis showed that inflammation marker (CD68) level was slightly lower in the hindbrains of *Galc*-iKO induced after P5 compared to those induced before P5; *n* = 6 per genotype. **e** Psychosine level was measured in the cervical spinal cords of the *Galc*-iKO, sampled when the mice were moribund (P60–85). Although psychosine levels were increased regardless of induction starting time, there was much less psychosine accumulation in the *Galc*-iKO ≥ P6 compared to the *Galc*-iKO ≤ P4, indicating of a correlation of psychosine level with the survival; *n* = 6 per genotype. **f** Further analysis of psychosine levels in specific CNS regions, including the cerebral cortex, cerebellum, and brainstem of both end-stage *Galc*-iKO ≤ P4 and *Galc*-iKO ≥ P6 brains, revealed that the early induction had more psychosine accumulation in the brainstem and spinal cord vs the cerebral cortex or cerebellum. The late-induced brains did not show any region-specific difference of psychosine level. The brainstem and spinal cord of the early-induced *Galc*-iKO (P3) had significantly higher psychosine levels than those of the late-induced *Galc*-iKO (P8), but the cerebral cortex and cerebellum did not; *n* = 3–6 per genotype. Data are presented as mean values ± SD. One-way ANOVA (**d**, **e**) and two-way ANOVA with Tukey's multiple comparison (**f**) tests were performed; *p < 0.05, **p < 0.01, and ***p < 0.001; ns not significant.

with swellings, breaks, or transections, suggesting severe structural disruption with axonal degeneration (Fig. 7e). To determine if the reduction in Thy1.1-YFP signal reflected neuronal/axonal damage secondary to demyelination and inflammation, presymptomatic mice were analyzed at P14 (Fig. 7f). Intriguingly, a

similar reduction of Thy1.1-YFP was seen in the brainstem of P14 *Galc*-KO mice, when clinical symptoms are normally absent. In line with previous studies, there was no evident microglial pathology at this early, "asymptomatic" timepoint (Fig. 7g–h and Supplementary Fig. 3b). Instead, we suspect that the reduced

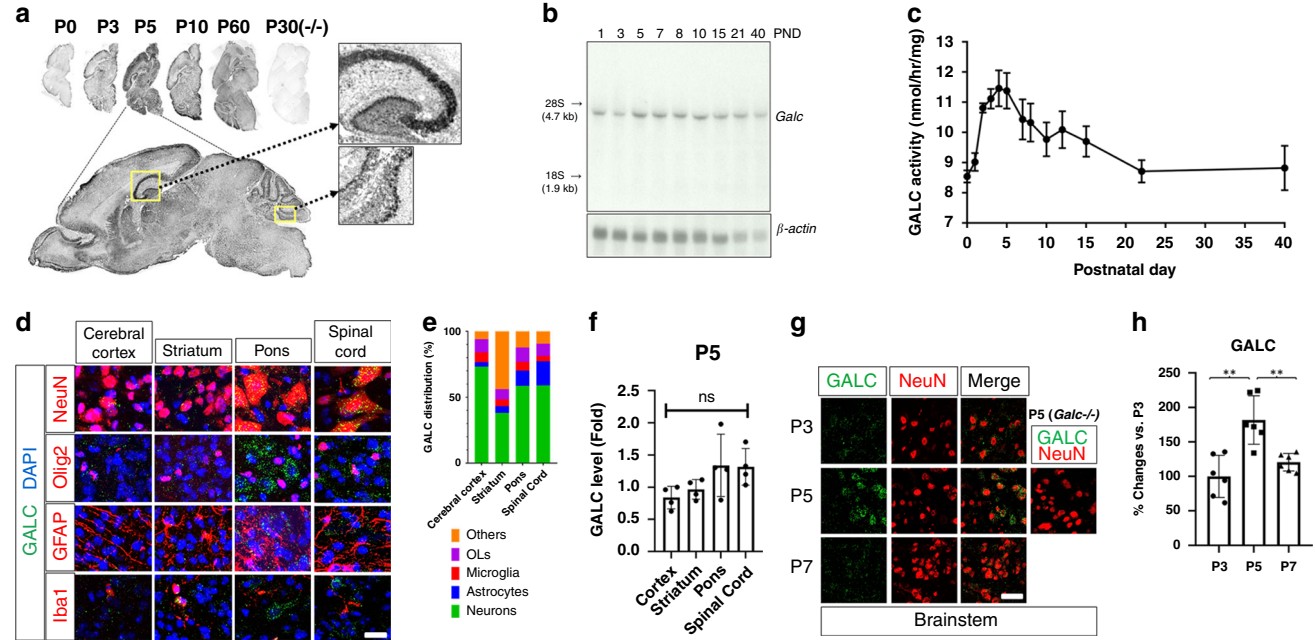

**Fig. 6 GALC levels reach their highest peak at P5 in the mouse brain. a** In situ hybridization and **b** northern blot analyses reveal that *Galc* transcript expression is low at P0, which then increase and reach its highest peak at P5 in WT brain. **c** GALC enzyme activity levels also show similar pattern of changes; $n = 6$ per age. **d** Immunohistochemistry on P5 brain shows that GALC protein is expressed in neurons (NeuN), microglia (Ibal), astrocytes (GFAP), and oligodendrocyte lineage cells (Olig2). Cell counts: 40–60 for NeuN, 20–30 for Olig2, 15–25 for GFAP, and 10 for Iba1 per image field. Six fields per region; $n = 3$. Scale bar = 50 μm. **e** Quantification of GALC in each cell type reveals that the majority of GALC at P5 is expressed in neurons (38–73%). **f** GALC level was similar among different brain regions, including cerebral cortex, striatum, pons, and spinal cord at P5. GALC level was represented by fold change compared to the value of cerebral cortex (cortex); $n = 4$. One-way ANOVA test was performed; ns not significant. **g**, **h** Confocal images of brainstem neurons double stained for GALC (green) and NeuN neuronal marker (red) in the P3–7 *Galc*[+/+] brain shows that GALC is highly expressed in NeuN-positive cells of the P5 brainstem. Scale bar = 20 μm; $n = 6$ per age. Error bars indicate SEM. Two-way ANOVA with Tukey's multiple comparison tests were used; **$p = 0.0016$ (P3 vs P5) and 0.0025 (P5 vs P7).

neuronal signal reflects a neural developmental perturbation in *Galc*-KO brains.

To determine if the attenuated neuronal signal was influenced by GALC expression in the early development, we crossed the same Thy1.1-YFP mice with our *Galc*-iKO system. We then analyzed brains of P35 mice that were either induced at P3 (*Galc*-iKO ≤ P4) or P7–8 (*Galc*-iKO ≥ P6; Fig. 8a). Comparing induction at both timepoints revealed that the brainstem, and in particular the pons, had a dramatic reduction of YFP neurons/axons in the *Galc*-iKO ≤ P4, when compared to the *Galc*-iKO ≥ P6 (Fig. 8b, c). Meticulous analysis of YFP signal intensity in serial sagittal brain sections validated that non-brainstem brain regions were not affected during the critical period (Supplementary Fig. 4). To confirm if this neuronal effect was the consequence of a developmental process, and not secondary to canonical KD pathogenesis, brains were promptly analyzed 24 h after the last tamoxifen administration (Fig. 8d). In line with a developmental defect, neuronal YFP signal in the brainstem was reduced immediately after tamoxifen induction in *Galc*-iKO ≤ P4 mice (Fig. 8e, f). Although a similar trend in YFP signal reduction was also observed by temporal *Galc* deletion starting at P7 (*Galc*-iKO ≥ P6), this finding was not statistically significant (Fig. 8f). Taken together, these results suggest that GALC expression at P5 is critical in the development and stability of brainstem neurons.

**Galc deficiency increases a population of immature brainstem neurons**. Although white matter and myelinating OL are thought to be the main contributors to Krabbe pathogenesis, they are not yet fully differentiated before P6. Furthermore, because neurons expressed the highest levels of GALC at P5 (Fig. 6), we explored

the effect of GALC on neuronal development in the brainstem. T-brain-1 (TBR1), a brain-specific T-box transcription factor, plays a critical role in the brain development. TBR1 expression is highest in immature neurons at the embryonic stage of brain development, and is gradually reduced as neurons mature[41]. Interestingly, the number and overall intensity of TBR1-positive cells were significantly increased in the brainstem of *Galc*-KO mice compared to WT during P3–7 (Fig. 9a–c), suggesting GALC may be involved in the maturation step of neurons from an immature stage. The same analysis in *Galc*-iKO mice showed that the brainstem of *Galc*-iKO ≤ P4 had a dramatic increase of TBR1-positive cell bodies and intensity compared to controls 24-h post induction (Fig. 9d–f). However, the TBR1 signals in the brainstem of *Galc*-iKO ≥ P6 did not show significant change, suggesting *Galc* deletion before P4 is critical for the persistent TBR1 expression.

The neuronal marker NeuN begins to be expressed during early embryogenesis in postmitotic neuroblasts, and remains expressed in differentiating and terminally differentiated neurons thereafter[42]. The overall NeuN-positive cell number was not different between WT and KO brainstems at P3, 5, and 7 (Supplementary Fig. 5a, b). This may be due to the fact that NeuN detects both mature neurons and immature postmitotic neurons during early brain development[43,44]. We also measured proliferating neurons by EDU labeling for 24-h prior to analysis. EDU-positive neurons colocalized with either TBR1 or NeuN were rare during the period, and were not different between WT and KO (Supplementary Fig. 5d, f). Since EDU labels cells only during the S phase of cell cycle, we also measured Ki67 signals which are expressed during all stages of cell proliferation including the G1, S, G2, and M phases of cell cycle. The Ki67 labeled cell number

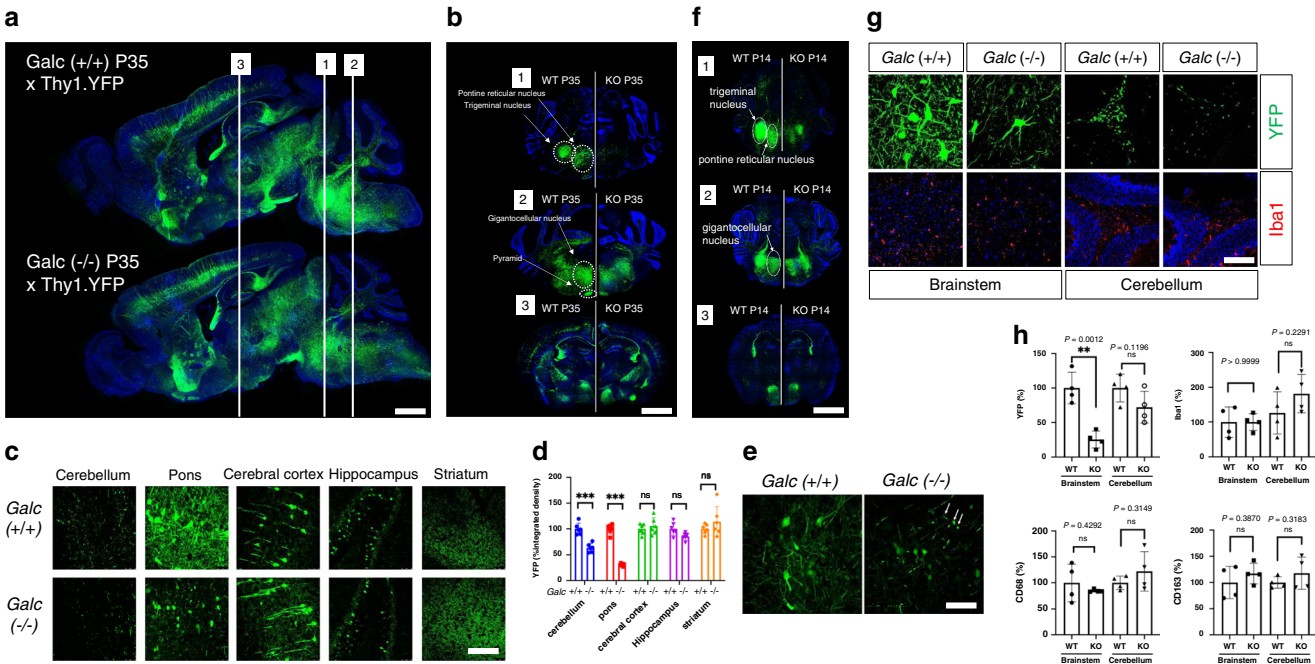

**Fig. 7 Brainstem is developmentally affected by GALC deficiency. a** Double transgenic mice of P35 *Galc*⁻/⁻; Thy1.1-YFP shows that a decrease in the density of YFP neurons/axons in the brainstem, compared to WT. **b** Coronal sections reveal pontine reticular/trigeminal/gigantocellular nuclei and pyramidal structures are dramatically affected. **c** Highly magnified images and quantification (**d**) show YFP signals significantly reduced in the cerebellum and pons (brainstem). Two-way ANOVA with Tukey's multiple comparison tests were used; $n = 6$ per genotype; ***$p = 0.0001$. **e** Mutant axons in *Galc*-KO appeared with swellings (arrow), break, or transections, indicating axonal degeneration. **f** P14 *Galc*⁻/⁻; Thy1.1-YFP brain has the same pattern of YFP reduction in trigeminal, pontine reticular, and gigantocellular nuclei, compared to WT. **g** Iba1-positive microglia are not activated in the brainstem and cerebellum of P14 *Galc*-KO, while YFP signals were much reduced. **h** Quantification of immunostained signals showed that microgliosis markers Iba1, CD68, and CD163 were not significantly changed, whereas YFP signals were much downregulated. Scale bars: **a** 2 mm; **b**, **f** 150 μm; **c**, **e**, **g** 50 μm; $n = 4$ per genotype. All data are presented as mean values ± SD. Unpaired Student's $t$ test was used; *$p < 0.05$, **$p < 0.01$, and ***$p < 0.001$, ns not significant.

was also unchanged in *Galc*-KO (Supplementary Fig. 5e, g), suggesting proliferation of neurons is not affected by *Galc* deficiency postnatally. While the number of neurons in the early developmental time period was not changed, there was a significant reduction in neurons of the P40 moribund *Galc*-KO brainstem (Supplementary Fig. 5c) presumably caused by apoptosis (Supplementary Fig. 5h, i), though it is not clear if this reduction is due to neuron-autonomous or secondary effects of other cell's pathogenesis.

**Galc deficiency causes reduced brainstem SOX2+Olig2+ OPCs.** During the critical period P4–6, GALC is expressed not only in neurons but also in other cell types, including OL-lineage cells (Fig. 6d, e). The murine pons, which is a core structure of the brainstem, quadruples in volume shortly after birth and peaks at P4, preceding myelination[45]. During this period, postnatal SOX2+Olig2+ OL-progenitor cells (OPC) expand 10–18-fold into the OL-lineage cells that later comprise the >90% of adult pons OLs. Therefore, in addition to neurons, it is possible that OPCs could be a key player affecting the development of KD brainstem during P4–6. To test this hypothesis, we counted OPC numbers in the brainstem of *Galc*-KO and WT at P3, P5, and P7 by immunostaining with the OL-lineage marker Olig2 and neural stem cell lineage marker SOX2. Interestingly, both numbers of SOX2+/Olig2+ and Olig2+ cells were dramatically reduced in the brainstem of the *Galc*-KO, compared to WT at all P3, 5, and 7 (Fig. 9g–i and Supplementary Fig. 6), suggesting that GALC has a specific role in the development of OLs in the postnatal brainstem. As before, we analyzed the number of OPCs in the brains of both *Galc*-iKO ≤ P4 and *Galc*-iKO ≥ P6. Interestingly, within 24 h of tamoxifen injection, SOX2+/Olig2+ and Olig2+ cell numbers

were dramatically reduced in the brainstems of both *Galc*-iKO ≤ P4 and *Galc*-iKO ≥ P6 (Fig. 9j–l and Supplementary Fig. 7). These data suggest that GALC expression is required for the expansion of the OPC population during brainstem development, but is independent of the critical period P4-6.

**Neuron-specific perinatal Galc influences neuronal differentiation without affecting OPCs.** Analysis of *Galc*-iKO highlights the importance of GALC in neurons during the early brainstem development. To determine if neuron-specific GALC ablation is sufficient to attenuate the maturation of neurons, we used pan-neuron-specific Thy1-Cre/ER^T2 mice[46] to induce neuron-specific temporal *Galc*-CKO mice. Although the Thy1-Cre/ER^T2 line is well characterized in the literature, we first confirmed Cre specificity and efficiency. To test both, tamoxifen was injected into Thy1-Cre/ER^T2;tdTomato starting at P2 tamoxifen at a dose of 25 μg per gram of body weight for four consecutive days. At P10 and P30, the animals were analyzed for the specificity of Cre expression and efficiency of Cre-loxP recombination (Supplementary Fig. 8a). The tdTomato protein is efficiently expressed in Thy1-positive cells in whole brain, including cerebellum, brainstem, cerebral cortex, and corpus callosum (Supplementary Fig. 8b, c). A total of 60–80% of Thy1 signals colocalized with tdTomato in the brain at both P10 and P30, indicating substantial specific recombination in neurons. Other cell markers Olig2 (OL), GFAP (astrocytes), and Iba1 (microglia) were minimally co-labeled with tdTomato, suggesting that Thy1 promoter-driven Cre expression is specific to neurons (Supplementary Fig. 8d). Next, we analyzed the population of immature neurons in the brainstem of Thy1-Cre/ER^T2-driven neuron-specific *Galc*-CKO induced starting at P2. As before, the

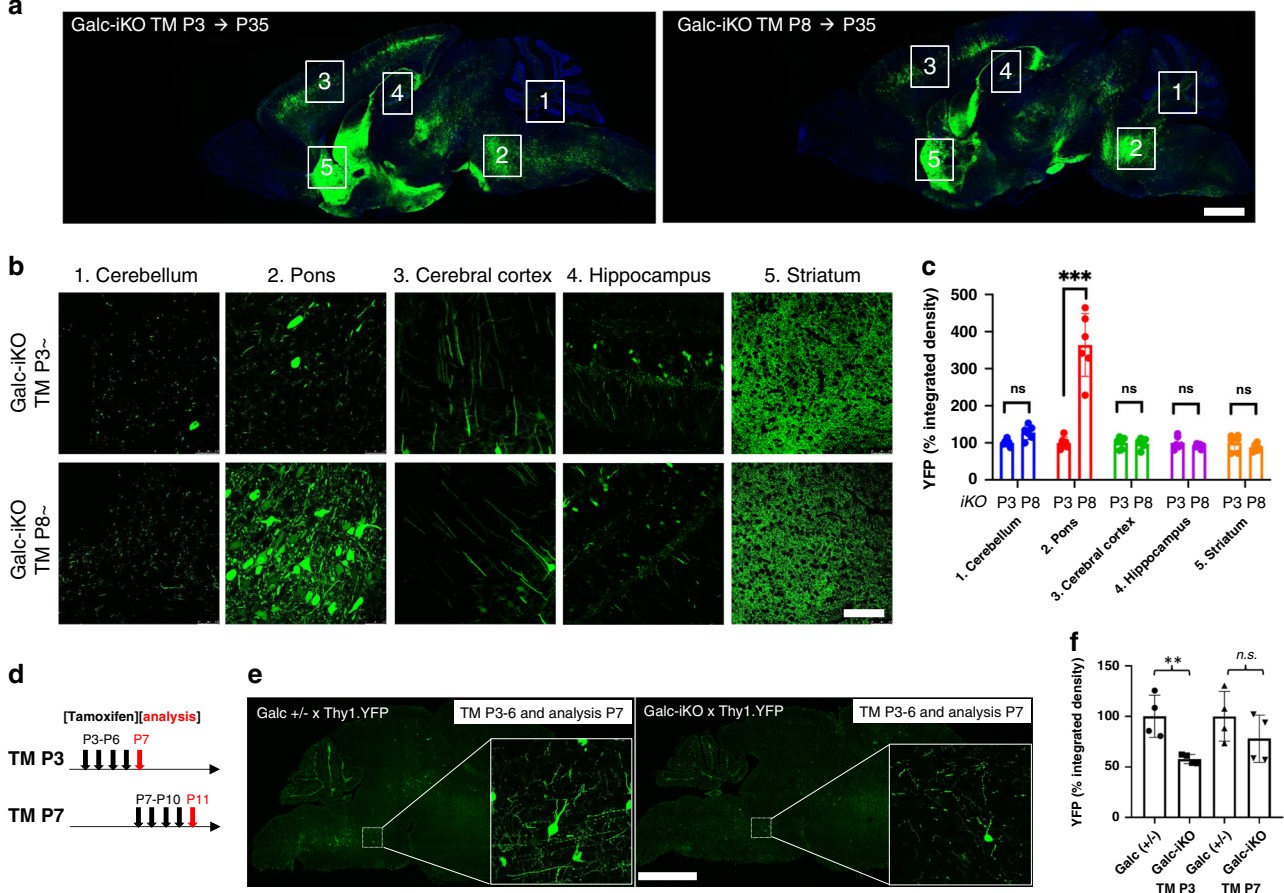

**Fig. 8 Brainstem development is affected during the critical period P4–6. a** *Galc*-iKO;Thy1.1-YFP shows a greater decrease in the density of YFP neurons/ axons in the brainstem when induced at P3, compared to the induction at P8. Highly magnified images (**b**) and quantification of YFP density (**c**) shows a significant change of signals in the pons of *Galc*-iKO (TM-P3) vs iKO (TM P8); $n = 5$ per genotype. *P* values are 0.8493 (cerebellum), 0.0001 (pons), and 0.9999 (others). **d**–**f** Monitoring of neuron-specific YFP signals 24 h after *Galc* deletion reveal a significant reduction of YFP intensity by induction at P3, but not by induction at P7. Scale bars: **a**, **e** 2 mm; **b** 50 μm; $n = 4$ per genotype. All data are presented as mean values ± SD. Two-tailed unpaired Student's *t* test was used; *$p < 0.05$, **$p < 0.01$, and ***$p < 0.001$; ns not significant.

analysis of mice at P60 showed an increase number of TBR1 cells and signal intensity in the brainstem of neuron-specific conditional mice vs WT (Fig. 10a–c). Similar to *Galc*-KO (Supplementary Fig. 5a, b), the overall NeuN-positive cell number was not different between WT and CKO brainstems at P60 (Fig. 10e). However, the number of Olig2-positive cells in the brainstem was not changed (Fig. 10d, f). These data suggest that neuronal GALC affects the maturation of neurons in a cell autonomous manner, but not the expansion of the OPC population.

We hypothesized that the brainstem psychosine accumulation secondary to perinatal *Galc* ablation (Fig. 5f) may correlate with the perturbation seen in brainstem neuronal differentiation. This hypothesis is in line with previous studies, which demonstrated that purified *Galc*-null granule neurons from *twitcher* had elevated psychosine levels[8]. We therefore used our neuron-specific inducible system, Thy1-Cre/ER[T2]-driven *Galc*-CKO mice induced at P3, and measured psychosine levels in P60 cerebral cortex, cerebellum, brainstem, and spinal cord. Interestingly, we found that psychosine did not accumulate secondary to neuron-specific *Galc* ablation (Supplementary Fig. 8e). This suggests that the observed maturation defect of brainstem neurons, which was dependent on neuron-specific GALC expression, was not directly influenced by psychosine accumulation. Furthermore, the accumulation of brainstem psychosine (Fig. 5f), was likely due to non-neuronal GALC ablation.

## Discussion

**Temporal requirement for GALC in KD pathogenesis**. To better understand the role of GALC in development, we used a conditional mutagenic approach and generated a novel mouse model of KD. We were particularly interested in the role of early postnatal GALC function, as empirical clinical evidence suggests that gene and stem cell therapy is more efficacious if delivered in a poorly defined presymptomatic window[47]. Our approach was to methodically induce *Galc* deletion at various postnatal time-points, thereby determining key postnatal events impacted by GALC deficiency. While we expected GALC to be important at a time period that coincided with myelination, we were surprised to find that GALC deficiency abruptly accelerated mortality when ablated before P4 compared to after P6 in C57BL6 mice. The protective role of perinatal GALC is particularly surprising and suggestive of a function unrelated to its canonical role in myelination.

This developmental role for GALC likely correlates with the observed clinical benefit of early, presymptomatic KD treatment. For example, presymptomatic infants transplanted before 1-month old have better outcomes than older presymptomatic infants[47]. KD pathogenesis may begin very early in development, perhaps even prenatally. In fact, Kobayashi et al. observed high level of psychosine accumulation in human fetal brain and spinal cord with KD, supporting the idea of prenatal disease

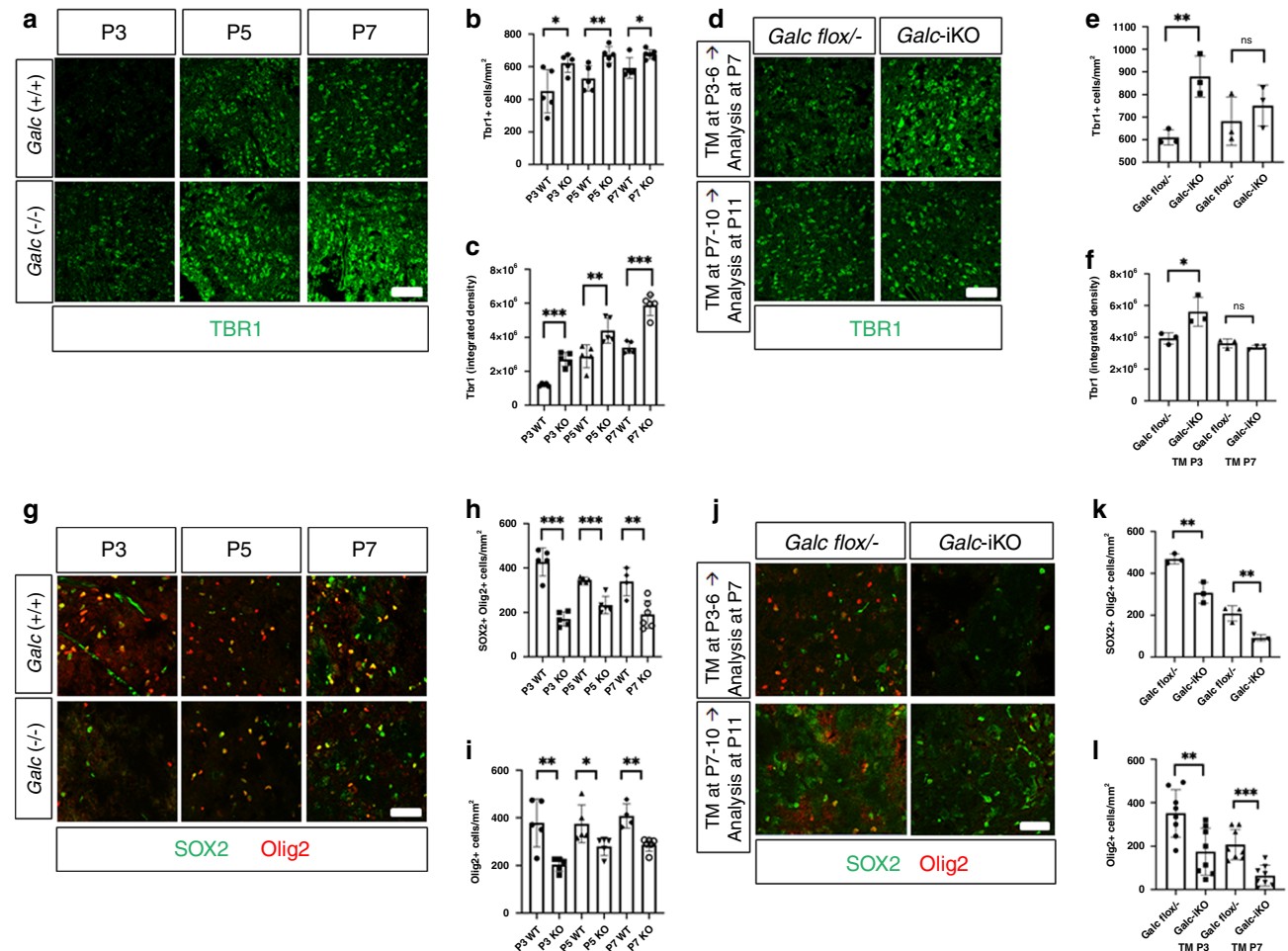

**Fig. 9 TBR1+ immature neurons are increased in the brainstem of *Galc* mutants, whereas SOX2+Olig2+ OPCs are reduced by GALC deficiency.**
**a** TBR1-positive cells were counted by immunohistochemistry on the brain sections of *Galc*-WT and KO at P3, P5, and P7. TBR1+ cells (**b**) and intensities (**c**) are significantly increased in *Galc*-KO in all ages; $n = 5$. $P$ values are 0.0286 (**b**, P3), 0.0080 (**b**, P5), 0.0170 (**b**, P7), 0.0001 (**c**, P3), 0.0097 (**c**, P5), and 0.0001 (**c**, P7). **d** Tamoxifen induction starting at P3 or P7 in *Galc*-iKO, and after 24 h TBR1-positive cells were stained. Quantification of TBR1+ cell numbers (**e**) and intensities (**f**) reveals *Galc* deletion at early time (P3) significantly increases TBR1+ cell number and intensity; however, the change of *Galc*-iKO brain induced at P7 was not significant; $n = 3$. $P$ values are 0.0086 (**e**, TM-P3), 0.4447 (**e**, TM-P7), 0.0409 (**f**, TM-P3), and 0.2185 (**f**, TM-P7). **g–i** SOX2 and Olig2-positive cells were counted by immunohistochemistry on the brain sections of *Galc*-WT and KO at P3, P5, and P7. In all three ages, the numbers of SOX2+Olig2+ and Olig2+ cells were reduced by *Galc*-KO; $n = 5$. $P$ values are 0.0001 (**h**, P3), 0.0003 (**h**, P5), 0.0067 (**h**, P7), 0.0057 (**i**, P3), 0.0427 (**i**, P5), and 0.0012 (**i**, P7). **j–l** Tamoxifen was administrated starting at P3 or P7 in *Galc*-iKO, and after 24 h SOX2 and Olig2-positive cells were counted. Both inductions significantly reduce the population of both SOX2+Olig2+ and Olig2+ cells, suggesting oligodendrocytes are affected by *Galc* deletion, but is independent of the critical period P4–6. Scale bar = 50 µm, $n = 3$ (**j**, **k**) and $n = 8$ (**l**). $P$ values are 0.0073 (**k**, TM-P3), 0.0070 (**k**, TM-P7), 0.0060 (**l**, TM-P3), and 0.0003 (**l**, TM-P7). All data are presented as mean values ± SEM. Two-way ANOVA with Tukey's multiple comparison tests were used; *$p < 0.05$, **$p < 0.01$, and ***$p < 0.001$; ns not significant.

development[48]. While the timescale between mice and humans is considerably different, the sequence of key events in brain maturation, such as neurogenesis, synaptogenesis, gliogenesis, and myelination between the two is consistent[49]. It was estimated that the mouse CNS at P1–3 corresponds to a gestational age of 23–32 weeks in humans, P7 to 32–36 weeks, and P10 to a term infant, at least in regards to white matter development[50]. Therefore, we anticipate that if our hypothesis on the early critical period of vulnerability is correct, then in utero treatments should have better outcomes than conventional postnatal treatment.

We also explored the possibility that the worsened clinical phenotype seen in early-induced mice reflected variable postnatal recombination. We were particularly concerned that the developing blood–brain barrier (BBB) in early neonatal mice would cause differences in tamoxifen uptake. Instead, we found

recombination was similar among our variable induction time-points (Fig. 3). This data fits with the fact that tight junctions between cerebral endothelial cells (the morphological basis for BBB impermeability) are functionally effective as soon as the first blood vessels penetrate the parenchyma of the developing brain[51], and also evidenced by analyzing tamoxifen metabolites in the postnatal brain[26]. Nonetheless, it is worth noting that the earliest induced *Galc*-iKO mice, occurring on postnatal day P0, still had significantly longer life spans (~P60) than either *Galc*-KO or *twitcher* (~P45; Fig. 1). This may indicate that the inducible Cre-loxP system is incomplete, and thereby delays progression of the KD phenotype[52]. Alternatively, deletion of *Galc* prior to birth may be required to fully recapitulate the global KD phenotype.

Another interesting and unexpected finding of our study was that residual GALC returned in the late course of the disease models (Fig. 3h and Supplementary Fig. 2). Our inducible

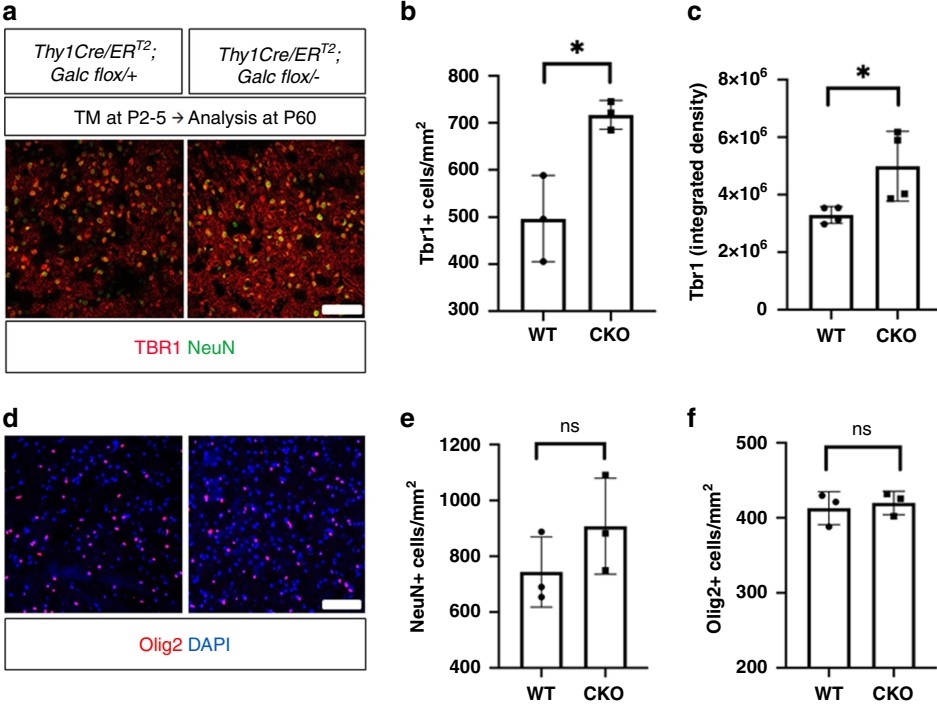

**Fig. 10 TBR1+ immature neurons are increased in the brainstem of neuron-specific *Galc* knockouts.** Immunostaining of TBR1 on the brainstem of Thy1-Cre/ER[T2]-driven *Galc* deletion induced starting at P2 (**a**) shows that the number of TBR1+ cells (**b**) and intensities (**c**) are significantly increased in CKO (Thy1-Cre/ER[T2];*Galc*[flox/−]) at P60, compared to WT (Thy1-Cre/ER[T2]; *Galc*[flox/+]). However, there were no changes in the numbers of NeuN (**e**) and Olig2 + cells (**d**, **f**). Scale bar = 50 μm; *n* = 3 per genotype. All data are presented as mean values ± SD. Unpaired two-tailed Student's *t* test was used; *$p < 0.05$; ns not significant. *P* values are 0.0166 (**b**), 0.0348 (**c**), 0.2536 (**e**), and 0.3682 (**f**).

Cre-LoxP system is dependent upon tamoxifen exposure for continued recombination events; thus, it seems plausible that substantial GALC activities could eventually return after *Galc* ablation especially if a non-recombined minority of cells repopulate over time. Intriguingly, despite the substantial GALC activity that returned in all *Galc*-iKO mice, the disease course remained unaltered and mice continued to deteriorate and inevitably reached moribundity (Fig. 2b, c). This finding may correlate with recent gene therapy trials for a number of LSDs, in which increased enzyme activity was detected at the whole parenchymal brain level, but unfortunately resulted in minimal clinical improvements[53,54]. This suggests that providing sufficient GALC to specific cell types at an appropriate temporal period is critical to treat KD.

**Cell and region-specific requirement for GALC in brain development.** After elucidating a developmental critical period that requires GALC, our next task was to discern the cellular, regional, and functional mechanism by which GALC is protective. Our work illustrates that GALC is not only required by myelinating cells, but is also expressed in many other brain cells as shown in Fig. 6. At the perinatal timepoint, GALC is most highly expressed in neurons, suggesting a neuron-autonomous role for GALC in early postnatal development. It is also known that GALC in neural stem cells maintains a subventricular zone neurogenic niche during the early postnatal period[55]. While the majority of neurons are generated embryonically, many neuronal populations must mature and differentiate postnatally. For example, the dentate gyrus (DG) is marked by substantial postnatal neuronal maturation, with a high number of immature granule cells (GCs) present during the first 2 weeks of postnatal life. These GC neurons then progress toward more mature patterns over the next few weeks of DG maturation[56]. Similarly,

postnatal refinement of neurons of the visual cortex is critical for the intrinsic properties, and plasticity needed for proper function and network activity[57]. Furthermore, most pontine circuits are postnatally acquired or refined[45], which contains nuclei that relay signals from the forebrain to the cerebellum, along with nuclei that deal primarily with vital functions[58]. Therefore, our finding that GALC influences postnatal neuronal differentiation, while surprising, is not unprecedented.

We show that brainstem development is dependent on intrinsic GALC expression by neurons. Specifically, brainstem neurons lacking GALC had perturbed differentiation and were in a more immature state. Neurons of this region ultimately had more axonal atrophy and degeneration, when *Galc* was deleted between P4–6. These pathological findings were not equal among all brain regions and were most pronounced in neurons of the brainstem. Interestingly, defective neurogenesis of the hindbrain and midbrain regions was previously noted in a zebrafish model of KD[59]. The brainstem was also previously implicated in KD as the region where pathology first develops[60]. While the functional consequence of brainstem dysfunction is difficult to directly assess, it is known to serve as an important relay between the forebrain, cerebellum, and other nuclei associated with vital functions[58]. In fact, most children who develop KD in infancy die before the age of 2, often from respiratory failure[61] and suspected autonomic dysfunction. In addition, even presymptomatic KD infants treated with HSCT continue to develop substantial motor impairments because of brainstem dysfunction involving corticospinal tract pathology[31]. Therefore, we suspect that GALC-dependent brainstem dysfunction may directly influence KD pathogenesis.

Despite our conclusion that P5 GALC influences brainstem neuronal differentiation, the molecular mechanism by which this happens remains unknown. Nonetheless, the argument for neuronal pathology caused by an autonomous role for neuronal

GALC is extensive. First and foremost, GALC is expressed by a diverse population of neurons in WT mice[38]. Secondly, neuronal pathology in KD precedes demyelination in both the CNS and PNS of *twitcher*, and increases in frequency with disease progression, ultimately leading to axonal damage and neuronal apoptosis[8,62]. In addition, neuronal pathology occurs in *twitcher* unable to synthesize galactosylceramide and psychosine, indicating a possible presence of a non-ceramide galactosyltransferase-based substrate for GALC in neurons[63]. Interestingly, this closely resembles a poorly characterized neuronal brainstem pathology in aged bone marrow transplanted *twitcher*[63]. Purified primary cultures of *twitcher* neurons were also able to produce psychosine directly, and had defects in abnormal neurites and disrupted axonal transport[64]. Finally, α-synuclein accumulated in the brains of *twitcher* and basal ganglia of Krabbe patients[65].

We found that prior to the onset of overt clinical symptoms, neuronal swellings, varicosities, and transected fibers were already detectable in *Galc*-deleted mutants (Figs. 7f–h and 8d–f). These findings preceded overt neuronal loss, as reductions in neuronal numbers only occurred shortly before death (Supplementary Fig. 5). This confirms previous studies, which suggest that axonal stress and dysfunction precede overt neuronal apoptosis in *twitcher*[8], a process which is consistent with the pathophysiology of other neurological disorders[66]. Since lysosomes are essential for synaptic biogenesis[67], it is conceivable that the degenerative process may be related to neuronal synaptic function at the axon terminal. Here, structural and functional defects may begin to impact synaptic efficiency, thereby causing overt neuronal toxicity[68], apoptosis, and ultimately overall brain development[69,70]. We suspect that immature brainstem neurons, influenced by autonomous GALC deficiency, are more vulnerable to toxins like psychosine, while also being less able to respond to local environmental signals, and are eventually eliminated[71].

We found that in the brainstem, expression of the immature neuron-specific transcription factor, TBR1, is developmentally regulated by GALC during the perinatal period. TBR1 is gestationally expressed in glutamatergic early-born cortical neurons[41,72]. However, beginning postnatally, TBR1 expression occurs in neurons of the thalamus and specific nuclei of the hindbrain including but not limited to the locus coeruleus, cerebellar nuclei, and Purkinje cells[72]. The role of TBR1 is therefore speculated to contribute to cellular differentiation as opposed to regional specification. In particular, it is hypothesized that the differentiation of radial glia to postmitotic projection neurons involve the sequential transition from PAX6 to TBR2 to TBR1 (ref. [73]). This sequential development has also been documented in neurons of the hindbrain, including the deep cerebellar nuclei[74]. The role of TBR1-expressing glutamatergic neurons of the hindbrain have not been fully clarified[75]. In addition to neuronal differentiation abnormalities, we observed that OL-lineage cell expansion was significantly reduced upon *Galc* deletion in the brainstem perinatally. Due to the common origin of neurons and OL-lineage cells, namely neural stem cells, it seemed conceivable that a dynamic interplay between the differentiation of OLs and neurons influenced proper brainstem development. However, our study showed that neuronal GALC did not influence the expansion of the OPC population during the critical period P4–6 (Fig. 10), suggesting a neuron-autonomous role of GALC for neuronal maturation exclusively. In fact, although myelination of the central nervous system appears normal in the early postnatal life of the *twitcher* mouse[16], axonal swellings, and varicosities occur as early as P7 (refs. [8,16]). These early axonal phenotypes, in the absence of demyelination, suggest a neuron-autonomous effect of GALC deficiency. Also, we cannot exclude the possibility that GALC may directly regulate the activation of TBR1 in the progression of neuronal maturation.

In summary, our findings highlight a previously unidentified critical period in which GALC is required for neuronal brainstem development. The sudden and abrupt change in the clinical phenotype of *Galc*-iKO mice induced before or after P5 suggests a highly dynamic developmental process related to the function of GALC. This finding is particularly interesting as P0–10 represents a presymptomatic timepoint in *Galc*-KO mice, occurring well before the majority of myelination in the murine brain. These findings are surprising, but may correlate with the presymptomatic therapeutic window seen in treating KD patients. Further studies are required to elaborate on the specific cellular mechanisms, which require GALC for brainstem development and if similar processes occur in other LSDs.

## Methods

**Generation of a conditional *Galc* floxed allele mouse**. A *Galc* targeting vector (Supplementary Fig. 1a) from a BAC clone harboring a mouse *Galc* gene (bMQ-165I15, Source Bioscience, UK) was made by using a genetic recombineering technique (GeneBridges GmbH, Germany). Of the 17 *Galc* exons, we decided to flank exon 9 by two loxP sites, because the region from exons 7–10 on the *Galc* gene is consistently expressed among all splice variants (Ensembl Genomes). Furthermore, removal of exon 9 leads to a frame shift mutation in the remaining protein coding transcript. The targeting vector was then injected into embryonic stem (ES) cells from 129S6 mice, ultimately yielding 225 surviving clones. PCR screening of the vector's short arm junction led to the selection of seven *Galc*-targeted positive clones, which were confirmed further by Southern blot (Supplementary Fig. 1b) and DNA sequencing of both arm junctions. Two positive ES cells were injected into blastocysts, and both cells generated chimeras successfully. One chimera transgene was germline transmitted (r in Supplementary Fig. 1a), which was crossed with Del-FLPe mice (JAX#012930)[76] to remove the FRT-neomycin-FRT cassette. The resultant floxed mice (f in Supplementary Fig. 1a) were mated to constitutive ubiquitous Cre (CMV-Cre; JAX#006054)[21] to validate if *Galc* can be completely deleted by Cre-loxP recombination (− in Supplementary Fig. 1a), and generate a null phenotype that would be comparable to *twitcher*. For genotyping of *Galc* floxed mouse, three primers were used; primer-1 = 5′-CATCATCCTGTTTCCACAGG-3′, primer-2 = 5′-AATATGTAGGGAGAGAGTGGTC-3′, and primer-3 = 5′-CTATTTTAAGGGAGTTCTGCCAGTG-3′. WT is 266 bp, loxP floxed is 393 bp, and null allele is 514 bp.

**Animals**. Experiments were conducted according to the protocols approved by the Institutional Animal Care and Use Committee of University at Buffalo and Roswell Park Cancer Institute. The housing condition was a 12-h light/12-h dark cycle at 23 °C with 50% humidity. All animals were maintained on the congenic background of C57BL/6 N. Breeder C57BL/6 N mice were purchased from Charles River (Wilmington, MA). CAG-Cre/ERᵀ(JAX#004682), CMV-Cre (JAX#006054), tdTomato (JAX#007905), Thy1.1-YFP (JAX#003782), and Thy1-Cre/ERᵀ² (JAX#012708) were purchased from The Jackson laboratory (Bar Harbor, ME). For Cre/ERᵀ-mediated recombination, a 5 mg/ml tamoxifen (Sigma-Aldrich) solution was prepared in autoclaved corn oil (Sigma-Aldrich). To achieve efficient *Galc* recombination by tamoxifen, multiple pilot experiments were conducted with varying doses (25–100 μg per gram of body weight) and times (2–5 consecutive days) of tamoxifen injection. Finally, perinatal CAG-Cre/ERᵀ; *Galc* floxed mice were injected intraperitoneally for four consecutive days (total 4×) with 25 μg/gram body weight, in pups between P0 and P10. This was the maximum achievable dosage while avoiding tamoxifen-induced gastric toxicity[77]. It has been reported that tamoxifen affects glucose/lipid metabolism[78] and myelination[79], which could theoretically affect the survival of GALC-deficient mice. To exclude this potential confounding factor, we treated *Galc*-KO mice with tamoxifen using the same paradigm, starting injections at P4. Importantly, tamoxifen did not affect the life span of *Galc*-KO mice (40–45 days, n = 3), arguing against the possibility of confounding tamoxifen toxicity. For proliferation assays, mice were pulsed with 100 μg per gram of body weight of 5-ethynyl-2′-deoxyuridine (Sigma-Aldrich) at 24 h before sacrifice by injection intraperitoneally.

**Tissue and immunohistochemistry**. Mice at defined ages were anesthetized, sacrificed, and then perfused with ice-cold phosphate-buffered saline (PBS) followed by 4% paraformaldehyde (PFA). Brains and spinal cords were dissected, postfixed in 4% PFA overnight, dehydrated in 30% sucrose at 4 °C, embedded in OCT (Leica), and processed as cryosections with a thickness of 20 μm. For immunohistochemistry, cryosections were permeabilized and blocked in blocking buffer (0.1% Triton X-100, 20% fetal bovine serum, and 2% bovine serum albumin in PBS) for 1 h at room temperature, and overlaid with primary antibodies overnight at 4 °C. Primary antibodies used were GALC[80] (1:500 dilution), NeuN (EMD Millipore; 1:200 dilution), Olig2 (Peprotech; 1:300 dilution), GFAP (Sigma-Aldrich and Abcam; 1:250 dilution, respectively), Iba1 (Wako; 1:200 dilution), CD68 (Bio-Rad; 1:200 dilution), CD163 (Bio-Rad; 1:200 dilution), tdTomato (Origene; 1:500

dilution), SOX2 (R&D systems; 1:300 dilution), TBR1 (Abcam; 1:100 dilution), cleaved caspase-3 (Cell Signaling Technology; 1:100 dilution), and Ki67 (Invitrogen; 1:200 dilution). After washing with PBS, sections were incubated with fluorophore-conjugated secondary IgGs (Alexa Fluor 488-AffiniPure F(ab′)2 Fragment Donkey anti-Chicken IgY, Alexa Fluor 488-AffiniPure Donkey Anti-Rat IgG, Cy3-AffiniPure Donkey anti-Goat IgG (H + L), Rhodamine-AffiniPure Donkey Anti-Rat IgG, Rhodamine Red-X-AffiniPure Donkey Anti-Rabbit IgG, Alexa Fluor 594-AffiniPure Goat anti-Mouse IgG2b, Alexa Fluor 594-AffiniPure Goat Anti-Mouse IgG-Fc subclass 2a, and Alexa Fluor 647-AffiniPure F(ab′)2 Fragment Donkey Anti-Rabbit IgG, all 1:800 dilution; from Jackson laboratories). After washing (3× for 5 min) with PBS, coverslips were mounted with Vectashield (Vector Laboratories) mounting medium and DAPI. For colorimetric cleaved caspase-3 staining, SignalStain® Boost IHC Detection Reagent (Cell Signaling Technology) was used with hematoxylin counterstain. Images were acquired and analyses performed while blinded to genotype. For the quantification of YFP, CD68/CD163, and Tuj1 levels, area and intensity of fluorescence were quantified with ImageJ[81]. Briefly, all the conditions were imaged with identical illumination, laser power, and gain parameters. The imaging and quantifications were performed in a blinded manner. The images were thresholded equally to achieve a binary image, subtracted by the background values obtained from the sections of non-YFP genotype or without primary antibody incubation, and then normalized to average of the control condition prior to statistical testing. The values are represented as % change compared to control.

**GALC quantification**. Specific cell type expression of GALC was performed by immunohistochemistry in matched sagittal brain sections. Specific brain regions were co-stained for GALC, cell-specific markers (NeuN for neurons; GFAP for astroglia; Olig2 for oligodendroglia; and Iba1 for microglia) and nuclear counterstaining (DAPI). Images were acquired using Lecia SP5 laser-scanning confocal microscopic analysis (Lecia Biosystems) with identical illumination, laser power, and gain parameters. Z-stacks were recorded utilizing sequential confocal images that were collected at 1 μm intervals covering 25 μm depth. The lowest threshold for the acquisition of GALC signal was always set by using the comparable Galc-KO sections. Mean fluorescent intensity was measured from six random, non-overlapping fields from at minimum three animals per genotype at each age, and then thresholding applied equally to each image to correct for background with ImageJ software (NIH). Maximum projection images were segmented using the autocontext classification with ilastik (v1.4b3)[82] for more accurate and unbiased quantification. The ilastik is a machine learning tool for image segmentation. To train the machine learning model three images of each region per cell stain were randomly selected for manual, sparse annotation of the different stains. The autocontext workflow requires this annotation twice, and uses the output from the first training as inputs to the second segmentation algorithm, resulting in improved segmentation of the cell stains from the background image. The segmented images of cellular regions were processed further with ImageJ to analyze the number and intensity of the GALC-stained vesicles per cell. Number of cell counts were 40–60 for NeuN, 20–30 for Olig2, 15–25 for GFAP, and 10 for Iba1 per image field.

**Transmission electron microscopy**. Mice were first anesthetized with 250 mg/kg body weight avertin, and then perfused with PBS and 2.5% gluta-aldehyde in phosphate buffer, and incubated in fixative for 1 week. After being postfixed, spinal cords were dissected and embedded in Epon. Ultrathin sections were cut and stained with uranyl acetate and lead citrate, and then collected on grids. The pictures were taken with a Tecnai electron microscope.

**Western blot analyses**. After homogenizing whole brains in RIPA buffer containing protease inhibitors (Roche) and PMSF, total protein extracts were separated by SDS–PAGE, transferred to PVDF membrane (Millipore), and blocked with 5% skim milk or BSA in TBS-Tween20. Primary antibodies used were GAPDH (Chemicon; 1:5000 dilution), β-tubulin (Novus Biologicals; 1:5000 dilution), tdTomato (Origene; 1:1000 dilution), TLR2 (R&D systems; 1:100 dilution), HA (Roche; 1:2000 dilution), and GFAP (Sigma-Aldrich; 1:1000 dilution). Peroxidase-conjugated secondary antibodies used were Anti-Rabbit IgG (H + L; Novus Biologicals; 1:2000 dilution), AffiniPure Donkey Anti-Chicken IgY, and AffiniPure Goat Anti-Rat IgG (all 1:3000 dilution, from Jackson laboratories). Specific protein bands were quantified by means of ImageJ and Image Studio (LI-COR Biosciences), and the values (in pixels) obtained were normalized on those of the corresponding β-tubulin bands. Normalized values were then expressed as the percentage of values obtained from region-matched bands of control WT tissues.

**GALC enzyme assay**. GALC activity was determined via competitive inhibition of β-galactosidase[2]. Snap-frozen whole brains were homogenized in 10 mM sodium phosphate buffer, pH 6.0, with 0.1% (v/v) Nonidet NP-40 by using Dounce homogenizer. A total of 1 μg of total brain lysates were mixed with 4-methylumbelliferone-β-galactopyranoside (final 1.5 mM) resuspended in 0.1/0.2 M citrate/phosphate buffer, pH 4.0, and AgNO₃ (final 55 μM) at 37 °C for 1 h. The enzymatic reactions were stopped by adding 0.2 M glycine/NaOH, pH 10.6. Fluorescence of liberated 4-ethylumbelliferone was measured on a spectrofluorometer (λex 360 nm; λem 446 nm).

**Northern blot hybridization**. Northern blot analyses with 20 μg of total RNAs from mouse brains were performed with NorthernMax kit (ThermoFisher Scientific)[83]. The probes were prepared as the [α-$^{32}$P] dCTP Galc exon fragment excised from the plasmid that harbors PCR product for exons 5–9 of mouse Galc (540 bp) and the mouse glyceraldehyde-3-phosphate dehydrogenase (508 bp).

**In situ hybridization**. Galc in situ hybridization was performed with a slight modification of the previous method[84]. Cryosections of brain were incubated with digoxigenin (DIG)-labeled antisense riboprobes for murine Galc. The probe was synthesized using T3 RNA polymerase (Promega) and labeled with DIG RNA label mix (Roche). An anti-DIG antibody conjugated with alkaline phosphatase (Roche) was used to probe sections, which were stained with 5-bromo-4-cloro-3-indlyl phosphate/nitro blue tetrazolium (Roche) chromogenic substrates.

**Measurement of psychosine**. Brain and spinal cord lysates were extracted in chloroform:methanol and partially purified on a strong cation exchanger column. After evaporation to dryness, each residue was dissolved in methanol and analyzed using LC–MS/MS[8].

**Statistical analyses**. Data collection and analysis were performed blind to the conditions of the experiments. Statistical analyses were performed using GraphPad Prism, version 8 (GraphPad Software, La Jolla, CA). The number or animals and cell cultures used for the experiments are indicated in the corresponding figure legends. Two-tailed unpaired Student's t test was used for the difference between two groups, except G-ratio analysis in which Welch's t test was used due to different sample number. One-way or two-way ANOVA with Bonferroni or Tukey's multiple comparisons were used for the differences among multiple groups depending on the number of variables. Values of $p < 0.05$ were considered to represent a significant difference. Data are presented as mean ± SEM or SD.

**Reporting summary**. Further information on research design is available in the Nature Research Reporting Summary linked to this article.

## Data availability

All other data are available from the corresponding authors upon reasonable request. The Galc floxed (flox/+) and null (+/−) mouse lines generated in this study will be deposited to JAX (www.jax.org) to be available to the scientific communitiy. Source data are provided with this paper.

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

## Acknowledgements

We would like to thank Aimee Stablewski of the Roswell Park transgenic animal facility for generation of the *Galc* conditional allele mouse, Christopher B. Eckman of the Biomedical Research Institute of New Jersey for providing the polyclonal GALC anti-body, and Ed Hurley of the Hunter James Kelly Research Institute for technical support of confocal and electron microscopic analyses. This work was supported by grants from the National Institutes of Health (R03-NS087359, R56-NS106023, and R01-NS112327 to D.S.), (R01-NS111715 to M.L.F. and L.W.), (F30-NS090835 to N.W.), (R01-NS065808 to E.R.B.), the Empire State Development Corporation for The Research Foundation—Krabbe Disease Research Working Capital (W753 to L.W.), the Empire State Development Corporation for Krabbe Disease Research Capital Equipment (U446 to L.W.); and a grant from Hunter's Hope Foundation.

## Author contributions

D.S. and N.W. designed and performed the experiments, analyzed data, made the figures, and wrote the paper, with input from L.W. and L.F. E.R.B. and D.N. analyzed psychosine level. J.F. and C.K. genotyped animals, and assisted image processing and quantifications. All authors reviewed and edited the paper.

## Competing interests

The authors declare no competing interests.
