## [Peer Review File · Nature Communications]

Reviewers' Comments:

Reviewer #1:

Remarks to the Author:

Krabbe disease (KD), also known as Globoid Cell Leukodystrophy, has been considered a prototypical demyelinating disorder since first described by Knud Krabbe in 1916. Kunihiro Suzuki and Yoshiyuki Suzuki identified deficiency of the lysosomal enzyme Galactocerebroside β -galactosidase (GALC) as being responsible for the disease. Kunihiro Suzuki, Lars Svennerholm and colleagues later discovered that in addition to the catabolism of the myelin-rich lipid galactosylceramide, galactosylsphingosine - present in very low quantities in normal nervous tissue but elevated in Krabbe disease - was also a substrate of GALC and a highly cytotoxic metabolite. Massive loss of myelin and myelin-producing cells is a well-established cardinal feature of the disease and consistent with the known role of GALC in lipid metabolism. However, in recent years there has been increasing recognition that myelin-independent autonomous neuronal abnormalities is part of KD pathology.

In this article, NI Weinstock and co-workers defined GALC temporal and spatial expression during mouse brain development. Using intricate genetic tools of GALC ablation, they identified the important role GALC plays in the maturation of T-box-brain-1 positive neurones during a narrow window of perinatal brainstem development. They concluded the absence of GALC during this short period is a critical contributor to the pathogenesis of KD. Although they have no current explanation as to the molecular mechanisms underlying this novel GALC neuronal function, the authors argue this finding might explain the necessity for early intervention to modify the course of human disease as well as that of animal models by different therapeutic modalities. Given that GALC appears to be expressed in a variety of neurones throughout the brain, it is intriguing the maturation deficit was observed predominantly in the brainstem. The findings presented in this elegant study extend our understanding of GALC function and identifies a possible reason for the failure in our attempts to treat this disease.

Minor critique

The manuscript is well illustrated but some of the figures are just too small given the amount of information provided.

The population sample for most figures is missing; please add this information for each figure.

The authors under "statistical analyses" tell us the error bars are SEM or SD, but it would be helpful to know when one or the other applies.

In figure 6, panel f, GALC staining is pseudo coloured green and NeuN cyan, but these colours are not distinguishable; please use a different palette of colours.

The manuscript is well written but there are some typos and grammar that needs addressing.

MB Cachon-Gonzalez

Reviewer #2:

Remarks to the Author:

NCOMMS-19-7936214 - Brainstem development requires galactosylceramidase and is critical for the pathogenesis of Krabbe Disease

Krabbe disease is a rare lysosomal storage disorder caused by a deficiency of galactosylceramidase (GALC). The major effect of GALC deficiency is the accumulation of psychosine in the CNS and PNS. The molecular mechanisms of Krabbe disease are not yet fully elucidated and a definite cure is still missing. In the infantile forms of the disease, demyelination and neurodegeneration start early and progress rapidly. Studies in animal models treated with gene/cell therapy as well as clinical observations in patients treated with HSCT highlight the need of early and robust GALC supply to delay the disease progression. Also, in vitro and in vivo studies suggest a psychosine-dependent primary neuronal damage contributing to the disease pathogenesis that may precede

the dysfunction/death of myelinating cells. Thus, the idea that GALC deficiency affects neurons/glia cells during the early postnatal CNS development is a long-standing theme in the Krabbe field and has been addressed previously.

In this study, the authors further explore this issue using a novel conditional Galc floxed mouse. They first generate global Galc-KO mice (by crossing Galc floxed mouse with CMV-Cre) showing a phenotype similar to that of the well-recognized twitcher mice. Then, by crossing Galc floxed mouse with an inducible ubiquitous Cre mice they generate a novel Galc-iKO mouse in which GALC ablation is induced by tamoxifen treatment at different postnatal days. They identify a critical temporal window between P0 and P4 in which Galc ablation results in a twitcher-like severe phenotype. Interestingly, Galc expression in brain tissues peaks at P5. Ablation performed >P6-P10 results in a milder phenotype and increased lifespan, correlating with lower psychosine accumulation. Finally, by crossing Galc-KO and Galc-iKO mice with Thy1.1-YFP mice, the authors show a specific effect of GALC deficiency in the brainstem neuronal population in the critical P0-P4 period, while the oligodendroglia cell population is affected by Galc ablation independently from this critical time window.

Overall, the rationale of the work is well explained, the mouse models are novel and relevant for the study, the experimental layout is sometime complex but supported by a detailed description and proper controls. The unexpected/counterintuitive results are highlighted and discussed. In my opinion, the last part of the work focusing on the brainstem defects is weaker as compared to the first part, and the conclusions drawn are a little too strong based on the data presented.

Specific points

1. Galc ablation at P0-P4 results in the most severe twitcher-like phenotype. Still, these mice survive longer (P60), possibly because of residual GALC activity (20% of normal detected at the latest time points) provided by few remaining WT cells. Is this unexpected enzymatic activity observed in tissues other than the brain (i.e. PNS, liver, spleen)?

"It was particularly intriguing that the residual GALC was unable to cross-correct affected tissues, as deduced from the severe phenotype". Indeed, there are no experimental data to support this statement.

"This result may correlate with deteriorative moribund pathology of HSCT-treated twitcher and KD patients despite the increased GALC activity (ref 25, 26, 27, 28), emphasizing the importance of providing GALC at a specific developmental period". Based on these results, GALC levels appear to be more or as critical as timing.

2. Data in Fig. 5e show similar levels of Psychosine in twitcher, Galc-KO, and Galc-iKO mice (P0-P4), despite the latter survive longer and show 20% of GALC enzymatic activity. This suggests that this level of GALC activity is not enough to ameliorate Psy storage in the spinal cord. Psy accumulation in CNS tissue of Twitcher mice follows a caudal to rostral pattern, showing the highest levels in the spinal cord (Ricca et al. HMG 2015). It would be relevant to check Psychosine levels in other CNS regions of the Galc-iKO mice, specifically in the brainstem, given the relevance this region is proposed to have in KD pathology

3. On the same line, GALC expression (Fig.6a-c) should be assessed in the CNS regions of interest (i.e. brainstem, cerebellum, spinal cord, etc) and not only in the whole brain. The images in Fig.6a are very difficult to interpret, a close up of specific regions would help to support the statements in the results: "Notably, the expression pattern of Galc transcripts by in situ hybridization suggest Galc may also be expressed in non-myelin regions of the P5 brain, particularly in neurons".

"The majority of GALC at P5 was expressed in neurons and was far higher than the expression in OL-lineage cells, astrocytes or microglia". Quantification of GALC immunoreactivity on IF stained

tissue slices is tricky, even using confocal images. Some more details regarding the settings (i.e. threshold to define negative vs. positive signal) and sample size (nr. of animals, nr. of slices/animal, nr. of fields/slice, total nr. of cells counted -neurons, astrocytes and oligos) would be appreciated. Which brain regions are shown in fig.6d and 6f? Are the proportions shown in fig.6e reproducible in different brain regions?

"Furthermore, neuronal GALC expression was developmentally influenced and peaked at P5 (Fig. 6f-g), similar to RNA and activity data (Fig. 6a-c). These results suggest a role for GALC in early neuronal development". It is not correct to compare RNA and activity data (obtained from whole brain tissue) with data in Fig. 6f-g, because they are not region-matched and (more importantly) cell type-matched (neurons account for 10% of the total cells in the brain).

4. The decreased density of YFP axons (Fig.7) and decreased nr. of NeuN+ cells (Fig.S3) suggest dysfunction/loss of neurons in the brainstem. However, the quantification of cleaved caspase-3+ cells and of the signal intensity are required to claim that "apoptotic cell death is significantly increased in the brain of muribond Galc-KO and Galc-iKO mice" and that "Galc-iKO \leq P4 had more prominent signals than Galc-iKO \geq P6" (legend to Fig. S3).

"High power neuronal magnification further emphasized the region specific nature of pathology seen in the Galc-iKO mice (Fig. 7c-d), ultimately suggesting that hindbrain pathology is likely a major consequence of KD pathogenesis". This is a strong claim based mainly on the analysis of YFP+ cells/axons. It would be nice to analyse additional neuronal markers (i.e. neurofilament) to support the result.

5. To my knowledge, TBR1 is expressed in the earliest-born cortical (glutamatergic) neurons from the preplate, being critical for cortical development. Are there evidences that this transcription factor has the same functional role in brainstem neurons, or whether TBR1 can be considered a general marker of immature neurons? Recognized markers of immature neurons should be tested, i.e. DCX, PSA-NCAM to validate the hypothesis of increased immature neurons due to early GALC deficiency in the brainstem. Also, these markers should be coupled with proliferation markers. What is the % of Ki67+ cells in the TBR1+ cell populations?

"The overall NeuN positive cell number was not different between WT and KO brainstems at P3, 5, and 7 (Supplementary Fig. 3a, b). This is probably due to the fact that NeuN detects both mature neurons and undifferentiated neuroepithelial cells". The authors should check the last sentence, or provide reference in support to it.

6. In Fig.8a, the YFP signal in the ventral region between the pons (2) and the striatum (5) as well in the region ventral to the hippocampus (4) is present in Galc-iKO mice TM P3->P35 but appears absent/reduced in mice ablated at P8. Is this due to the non-identical sagittal plane of the representative tissue slices? (details on the YFP quantification method are needed). Is it worth considering (and testing) the possibility that GALC deficiency affects differently neuronal populations in different CNS areas at different developmental times?

Overall, the results presented here support the role of GALC to ensure proper development/morphology/survival of brainstem neurons in a cell-autonomous manner; however, they do not completely exclude a similar role in neurons of other brain regions (more subtle differences – dysfunction vs. cell death- may have been missed in this analysis). In particular, a comparison with more caudal CNS regions (i.e. spinal cord) that are normally highly and early affected in KD and express the highest psychosine load, would be needed to support the main claim of the paper.

Also, the link between the neuronal GALC deficiency and psychosine in this model is still unclear. In this perspective, the comparative assessment of GALC activity and psychosine in selected CNS regions as well as the measurement of psychosine in tdTomato-Thy1+ neurons isolated from

brainstem, spinal cord and more rostral regions would help clarifying this link

These experiments led to the identification of a critical perinatal time window in which the developing murine brain is most vulnerable to GALC deficiency. As the author discuss, this period roughly corresponds (in terms of brain development) to the 2nd-3rd trimester of human gestation, further suggesting that in KD CNS pathogenesis starts before birth, possibly affecting neurogenesis and/or the function of early post-mitotic neurons and oligodendroglial progenitors. The brainstem is apparently most affected (even if the reason for this apparent selectivity is not elucidated) but other brain regions may be involved, although at lesser extent or at a different stage of CNS development.

Minor points

Discussion

"KD pathogenesis may begin very early in development, perhaps even prenatally". There are indications of psychosine accumulation in human fetal brain tissues (Igisu, Science 1984)

"This finding may correlate with recent gene therapy trials for a number of LSDs in which increased enzyme activity was detected at the whole parenchymal brain level, but unfortunately resulted in minimal clinical improvements(ref 43, 44). This suggests that providing GALC in specific developmental period is critical to treat KD". Providing sufficient levels of functional enzyme is as critical as the timing at which GALC is provided. Results of this study further suggest that >20% of normal levels (as measured in total brain lysates) are likely needed to prevent accumulation/reduce storage of psychosine in brain tissues, even if region-specific differences have to be considered. This should be better investigated.

"We show that brainstem development is dependent on intrinsic GALC expression by neurons. Specifically, brainstem neurons lacking GALC had perturbed differentiation and were in a more immature state". This statement is not fully supported by the results presented here (see point 5-6)

Methods and figure legends

- Please indicate the statistical test used, the sample size, and the significance of the statistical comparisons for each graph/set of data in all the figure legends.
- Cell counts and z-stacks (methods): please add details (see point 3)
- Add method for YFP quantification (see point 6)

Manuscript # NCOMMS-19-7936214A

Reviewer #1 (Remarks to the Author):

Krabbe disease (KD), also known as Globoid Cell Leukodystrophy, has been considered a prototypical demyelinating disorder since first described by Knud Krabbe in 1916. Kunihiko Suzuki and Yoshiyuki Suzuki identified deficiency of the lysosomal enzyme Galactocerebroside β -galactosidase (GALC) as being responsible for the disease. Kunihiko Suzuki, Lars Svennerholm and colleagues later discovered that in addition to the catabolism of the myelin-rich lipid galactosylceramide, galactosylsphingosine - present in very low quantities in normal nervous tissue but elevated in Krabbe disease - was also a substrate of GALC and a highly cytotoxic metabolite. Massive loss of myelin and myelin-producing cells is a well-established cardinal feature of the disease and consistent with the known role of GALC in lipid metabolism. However, in recent years there has been increasing recognition that myelin-independent autonomous neuronal abnormalities is part of KD pathology. In this article, NI Weinstock and co-workers defined GALC temporal and spatial expression during mouse brain development. Using intricate genetic tools of GALC ablation, they identified the important role GALC plays in the maturation of T-box-brain-1 positive neurones during a narrow window of perinatal brainstem development. They concluded the absence of GALC during this short period is a critical contributor to the pathogenesis of KD. Although they have no current explanation as to the molecular mechanisms underlying this novel GALC neuronal function, the authors argue this finding might explain the necessity for early intervention to modify the course of human disease as well as that of animal models by different therapeutic modalities. Given that GALC appears to be expressed in a variety of neurones throughout the brain, it is intriguing the maturation deficit was observed predominantly in the brainstem. The findings presented in this elegant study extend our understanding of GALC function and identifies a possible reason for the failure in our attempts to treat this disease.

We sincerely thank the reviewer for recognizing the novelty and rigor of our study.

Minor critique

The manuscript is well illustrated but some of the figures are just too small given the amount of information provided.

We agree that some figures are too small. We adequately edited all figures to be more legible as suggested.

The population sample for most figures is missing; please add this information for each figure.

The population information has been added in all figure legends, as suggested.

The authors under “statistical analyses” tell us the error bars are SEM or SD, but it would be helpful to know when one or the other applies.

We specified SEM or SD for all error bars in the corresponding figure legends, as suggested.

In figure 6, panel f, GALC staining is pseudo coloured green and NeuN cyan, but these colours are not distinguishable; please use a different palette of colours.

We agree and now show the GALC channel with green and cell markers with red in Fig. 6f.

The manuscript is well written but there are some typos and grammar that needs addressing.

The revised manuscript has been carefully proofread and edited accordingly, as suggested.

Reviewer #2 (Remarks to the Author):

NCOMMS-19-7936214 - Brainstem development requires galactosylceramidase and is critical for the pathogenesis of Krabbe Disease

Krabbe disease is a rare lysosomal storage disorder caused by a deficiency of galactosylceramidase (GALC). The major effect of GALC deficiency is the accumulation of psychosine in the CNS and PNS. The molecular mechanisms of Krabbe disease are not yet fully elucidated and a definite cure is still missing. In the infantile forms of the disease, demyelination and neurodegeneration start early and progress rapidly. Studies in animal models treated with gene/cell therapy as well as clinical observations in patients treated with HSCT highlight the need of early and robust GALC supply to delay the disease progression. Also, in vitro and in vivo studies suggest a psychosine-dependent primary neuronal damage contributing to the disease pathogenesis that may precede the dysfunction/death of myelinating cells. Thus, the idea that GALC deficiency affects neurons/glia cells during the early postnatal CNS development is a long-standing theme in the Krabbe field and has been addressed previously.

While we agree that previous studies have suggested a role for psychosine toxicity in neuron and glial cells during early postnatal CNS development, we respectfully disagree with the argument that this concept “is a long-standing theme in the Krabbe field that has been addressed previously”. In fact, there is no clear answer to the main three

aspects of this long-standing argument: 1) if pathology happens very early, 2) why neurons may be affected independently of demyelination and 3) is there a role of GALC deficiency independent from psychosine toxicity. For point 1, the major evidence for a role of GALC in early postnatal development stems from empirical evidence that attempts to rescue KD pathology in the twitcher mice and KD patients were improved when therapies are delivered early. While this point is a strong indication of an underlying role for GALC in perinatal development, the reason is unknown and others argued that this phenomenon highlighted technical issues with therapeutic interventions, which were more effective when delivered early (H. Allewelt et al., 2018; H. B. Allewelt et al., 2016; Rafi, Rao, Luzi, Curtis, & Wenger, 2012). Point number 2, that neurons may be affected early and independently from demyelination, is controversial and have been contested by prominent scientists in the field, who argue that neurons are affected secondarily to psychosine accumulation generated by oligodendrocytes during demyelination (Suzuki, 2004). The strongest piece of evidence of an early neuronal role came from Castelvetti et al., a landmark paper that highlighted the role of neurons in KD (Castelvetti et al., 2011). This study found the surprising data that axonal swellings occurred as early as P7 in the twitcher mouse. While these studies were striking and showcased a neuronal basis of KD pathogenesis, they could not draw a series of causal events because they were descriptive and occurred in the twitcher mice, in which mutant GALC was absent ubiquitously from birth. Therefore, correlating the early observed axonal morphologic pathology, with overall ramifications to clinical disease and functional consequences, was not possible due to the progressive nature of GALC ablation in all cell types. Furthermore, because GALC was absent in all cell types, the phenotype observed could be caused from GALC loss-of-function independently in neurons, glia, or a combination of both cell types. 3) Finally, these previous studies focused on psychosine toxicity, not of a possible GALC role in neuronal development independent of psychosine. Thus, much remains unknown regarding the role of GALC during this pre-myelination developmental time period. Our study is the first to compare the in vivo clinical course of GALC ablation early or late in postnatal development, a feat that was previously not possible. Furthermore, we now use a neuron-specific ablation method to show that these consequences occur, at least in part, by Galc autonomous loss of function in neurons.

In this study, the authors further explore this issue using a novel conditional Galc floxed mouse. They first generate global Galc-KO mice (by crossing Galc floxed mouse with CMV-Cre) showing a phenotype similar to that of the well-recognized twitcher mice. Then, by crossing Galc floxed mouse with an inducible ubiquitous Cre mice they generate a novel Galc-iKO mouse in which GALC ablation is induced by tamoxifen treatment at different postnatal days. They identify a critical temporal window between P0 and P4 in which Galc ablation results in a twitcher-like severe phenotype.

Interestingly, Galc expression in brain tissues peaks at P5. Ablation performed >P6-P10 results in a milder phenotype and increased lifespan, correlating with lower psychosine accumulation. Finally, by crossing Galc-KO and Galc-iKO mice with Thy1.1-YFP mice, the authors show a specific effect of GALC deficiency in the brainstem neuronal population in the critical P0-P4 period, while the oligodendroglia cell population is affected by Galc ablation independently from this critical time window.

Overall, the rationale of the work is well explained, the mouse models are novel and relevant for the study, the experimental layout is sometime complex but supported by a detailed description and proper controls. The unexpected/counterintuitive results are highlighted and discussed. In my opinion, the last part of the work focusing on the brainstem defects is weaker as compared to the first part, and the conclusions drawn are a little too strong based on the data presented.

We thank the reviewer for emphasizing the importance of this study in the KD field. We agree and added new data in Fig. 5f and Supplementary Fig. 3-5, to support and expand on the brainstem defects caused by GALC deficiency.

Specific points

1. Galc ablation at P0-P4 results in the most severe twitcher-like phenotype. Still, these mice survive longer (P60), possibly because of residual GALC activity (20% of normal detected at the latest time points) provided by few remaining WT cells. Is this unexpected enzymatic activity observed in tissues other than the brain (i.e. PNS, liver, spleen)?

To answer this question, in this revised manuscript we measured GALC activities in other peripheral organs including sciatic nerve, liver and spleen, which showed that substantial GALC activity had returned in the sciatic nerve of the moribund Galc-iKO animals, with modest return of activity in the liver, and minimal GALC activity in the spleen (Fig. S2), indicating unique tissue-specific return of GALC activity in moribund Galc-iKO mice.

“It was particularly intriguing that the residual GALC was unable to cross-correct affected tissues, as deduced from the severe phenotype”. Indeed, there are no experimental data to support this statement.

We thank the reviewer for the comment and agree that our argument is not fully supported by the previously provided data. Our intent was not to prove that cross-correction of GALC does not happen in our model, but rather to emphasize that the moribund KD phenotype existed despite the remaining GALC. We rewrote the text to clarify this point, including relevant recent citations: We theorize that the lack of phenotypic rescue via residual GALC may be explained by GALC levels that are still too

low, expressed in the wrong cell type, or present at the wrong time period. We have explored cross-correction in another project that has been recently published (Weinstock et al., 2020). This paper and other shows that, cross-correction of GALC and other lysosomal hydrolases (Wolf et al., 2020) is not efficient in vivo, and therefore the function of GALC may be restricted to the subset of cells that directly express it.

“This result may correlate with deteriorative moribund pathology of HSCT-treated twitcher and KD patients despite the increased GALC activity (ref 25, 26, 27, 28), emphasizing the importance of providing GALC at a specific developmental period”. Based on these results, GALC levels appear to be more or as critical as timing. *We agree that current data does not exclude the possibility that, in addition to timing of GALC expression, a certain minimum threshold of GALC activity may be required to rescue the phenotype. We edited the contents appropriately: Taken together, multiple observations from different studies (Hawkins-Salsbury et al., 2015; Marshall et al., 2018; Reddy et al., 2011; Wright, Poe, DeRenzo, Haldal, & Escolar, 2017) including this manuscript emphasize the nuance of providing sufficient GALC to specific cell types at an appropriate temporal period.*

2. Data in Fig. 5e show similar levels of Psychosine in twitcher, Galc-KO, and Galc-iKO mice (P0-P4), despite the latter survive longer and show 20% of GALC enzymatic activity. This suggests that this level of GALC activity is not enough to ameliorate Psy storage in the spinal cord. Psy accumulation in CNS tissue of Twitcher mice follows a caudal to rostral pattern, showing the highest levels in the spinal cord (Ricca et al. HMG 2015). It would be relevant to check Psychosine levels in other CNS regions of the Galc-iKO mice, specifically in the brainstem, given the relevance this region is proposed to have in KD pathology

We thank the reviewer for this remark, that promoted more experiments that now better correlate with the difference in severity observed during early or late Galc deletion. We agree that the differential accumulation of psychosine could affect brain region-specific progression of the disease. As suggested by the reviewer, in this revised manuscript we measured psychosine in the cerebral cortex, cerebellum, and brainstem of both end-stage Galc-iKO_{≤P4} (P3) and Galc-iKO_{≥P6} (P8) brains, to investigate if the timing of Galc deletion affects region-specific accumulation of psychosine. Interestingly, the early induction (P3) had more psychosine accumulation in the brainstem and spinal cord as compared to cerebral cortex or cerebellum. In contrast, the late induced brains (P8) did not show any region-specific difference of psychosine level (Fig. 5f). Furthermore, the brainstem and spinal cord of the early induced Galc-iKO (P3) had significantly higher psychosine level than those of the late induced Galc-iKO (P8), but the cerebral cortex and cerebellum did not. These data suggest, as the reviewer predicted, that early Galc

deletion results in more psychosine accumulation in the brainstem and spinal cord than other brain regions.

3. On the same line, GALC expression (Fig.6a-c) should be assessed in the CNS regions of interest (i.e. brainstem, cerebellum, spinal cord, etc) and not only in the whole brain. The images in Fig.6a are very difficult to interpret, a close up of specific regions would help to support the statements in the results: “Notably, the expression pattern of Galc transcripts by in situ hybridization suggest Galc may also be expressed in non-myelin regions of the P5 brain, particularly in neurons”.

Fig. 6a has been modified to show a close up of various CNS regions including hippocampal and cerebellar regions. We also assessed the region-specific GALC level in cerebral cortex, striatum, pons, and spinal cord at P5, which showed that the overall expression of GALC protein was similar among different CNS regions (Fig. 6f).

“The majority of GALC at P5 was expressed in neurons and was far higher than the expression in OL-lineage cells, astrocytes or microglia”. Quantification of GALC immunoreactivity on IF stained tissue slices is tricky, even using confocal images. Some more details regarding the settings (i.e. threshold to define negative vs. positive signal) and sample size (nr. of animals, nr. of slices/animal, nr. of fields/slice, total nr. of cells counted -neurons, astrocytes and oligos) would be appreciated. Which brain regions are shown in fig.6d and 6f? Are the proportions shown in fig.6e reproducible in different brain regions?

Detailed methods of GALC quantification was amended in the methods section and Fig. 6 legend. Specific cell type expression of GALC was performed by immunohistochemistry in matched sagittal brain sections. Specific brain regions were co-stained for GALC, cell specific markers (NeuN for neurons; GFAP for astroglia; Olig2 for oligodendroglia; Iba1 for microglia) and DAPI counterstaining. Images were acquired using Lecia SP5 laser-scanning confocal microscopic analysis. Z-stacks were recorded utilizing sequential confocal images that were collected at 1 μ m intervals covering 25 μ m depth. The lowest threshold for the acquisition of GALC signal was always set by using the comparable Galc-KO sections. Mean fluorescent intensity was measured from six random, nonoverlapping fields from at minimum three animals per genotype at each age, and then thresholding applied equally to each image to correct for background with ImageJ software (NIH). Maximum projection images were segmented using the auto context classification with ilastik (v1.4b3)(Berg et al., 2019). The segmented images were processed further with ImageJ to analyze the number and intensity of the GALC-stained vesicles per cell. Number of cell counts were 40-60 for NeuN, 20-30 for Olig2, 15-25 for GFAP, and 10 for Iba1 per image-field. The original brain region was only brainstem (pons). In this revised manuscript, we have also analyzed other brain regions including cerebral cortex, striatum, and spinal cord, and added in Fig. 6d-f. The new

data shows that the proportions of GALC distribution was different depending on brain regions. Interestingly, neurons have the most GALC expression in all brain regions (Fig. 6e).

“Furthermore, neuronal GALC expression was developmentally influenced and peaked at P5 (Fig. 6f-g), similar to RNA and activity data (Fig. 6a-c). These results suggest a role for GALC in early neuronal development”. It is not correct to compare RNA and activity data (obtained from whole brain tissue) with data in Fig. 6f-g, because they are not region-matched and (more importantly) cell type-matched (neurons account for 10% of the total cells in the brain).

We agree with the reviewer that our original data inappropriately compared overall RNA and activity to protein level without specifying region and cell types. To carefully determine the cellular expression of GALC protein in perinatal brain development, in this revised manuscript we performed GALC colocalization experiments using various cell markers throughout multiple anatomic regions in the P5 brain. GALC protein was expressed in all brain cell types including neurons, microglia, astrocytes, and OL lineage cells (Fig. 6d-e). The majority of GALC at P5 was expressed in neurons (38-73%) and was far higher than the expression in OL-lineage cells (7-11%), astrocytes (5-18%) or microglia (4-7%). The unlabeled GALC-positive population, ‘others’ (6-44%), presumably reflect undifferentiated cell types (Weyer & Schilling, 2003). However, this new GALC protein data is still not adequate enough to directly compare to region and cell specific Galc transcript levels. Therefore, the corresponding paragraph has been edited accordingly to more accurately reflect the findings in context without ‘similar to RNA and activity data’ as: Furthermore, neuronal GALC expression was developmentally influenced and peaked at P5 (Fig. 6g, h). These results suggest a role for GALC in early neuronal development, in the result section.

4. The decreased density of YFP axons (Fig.7) and decreased nr. of NeuN+ cells (Fig.S3) suggest dysfunction/loss of neurons in the brainstem. However, the quantification of cleaved caspase-3+ cells and of the signal intensity are required to claim that “apoptotic cell death is significantly increased in the brain of moribund Galc-KO and Galc-iKO mice” and that “Galc-iKO_≤P4 had more prominent signals than Galc-iKO_≥P6” (legend to Fig. S3).

Detailed quantification of the percentage of apoptotic cells (that was counted by cleaved caspase-3 positive cells per total nuclei) and statistics have been added in Fig. S5h-i, as suggested.

“High power neuronal magnification further emphasized the region specific nature of pathology seen in the Galc-iKO mice (Fig. 7c-d), ultimately suggesting that hindbrain pathology is likely a major consequence of KD pathogenesis”. This is a strong claim

based mainly on the analysis of YFP+ cells/axons. It would be nice to analyse additional neuronal markers (i.e. neurofilament) to support the result.

We further assessed the level of Tuj1, a well-known neuronal processes marker. The data show that Tuj1 is less expressed in the brainstem of P35 moribund Galc-KO, compared to WT (Fig. S3a).

5. To my knowledge, TBR1 is expressed in the earliest-born cortical (glutamatergic) neurons from the preplate, being critical for cortical development. Are there evidences that this transcription factor has the same functional role in brainstem neurons, or whether TBR1 can be considered a general marker of immature neurons? Recognized markers of immature neurons should be tested, i.e. DCX, PSA-NCAM to validate the hypothesis of increased immature neurons due to early GALC deficiency in the brainstem. Also, these markers should be coupled with proliferation markers. What is the % of Ki67+ cells in the TBR1+ cell populations?

In this revised manuscript, we have counted Ki67+ TBR1+ double positive neurons in the brainstem, which cells were minimally present during the age P3-7 and not different significantly by GALC deficiency (Fig. S5d-e), indicating proliferation of neurons is not affected by Galc deficiency postnatally. We also performed immunohistochemistry with commercially available antibodies against DCX (Abcam #ab18723) and PSA-NCAM (EMD Millipore #Mab5324). In our hand, it was not possible to count single DCX+ or PSA-NCAM+ discrete cells in the brainstem, because they are mostly expressed in the cytoskeleton or plasma membrane (see figure on the right). However,

we did observe more positive signals by both antibodies in Galc-KO vs. WT, consistent with the hypothesis that there are more immature neurons in the brainstem of Galc-KO mice. Scale bar=50 μ m. We have also amended on the role of TBR1 in the discussion section: TBR1 is gestationally expressed in glutamatergic early-born cortical neurons (Bulfone et al., 1995; Hevner et al., 2001). However, beginning postnatally, TBR1 expression occurs in neurons of the thalamus and specific nuclei of the hindbrain including but not limited to the locus coeruleus, cerebellar nuclei and Purkinje cells (Bulfone et al., 1995). The role of TBR1 is therefore speculated to contribute to cellular differentiation as opposed to regional specification. In particular, it is hypothesized that the differentiation of radial glia to postmitotic projection neurons involve the sequential transition from PAX6 to TBR2 to TBR1 (Englund et al., 2005). This sequential development has also been documented in neurons of the hindbrain, including the deep

cerebellar nuclei (Fink et al., 2006). The role of TBR1-expressing glutamatergic neurons of the hindbrain have not been fully clarified (Hoshino, 2012).

“The overall NeuN positive cell number was not different between WT and KO brainstems at P3, 5, and 7 (Supplementary Fig. 3a, b). This is probably due to the fact that NeuN detects both mature neurons and undifferentiated neuroepithelial cells”. The authors should check the last sentence, or provide reference in support to it.

We added the corresponding references. Indeed, it has been known that NeuN can detect both mature neurons and immature post-mitotic neurons during early brain development (Mullen, Buck, & Smith, 1992; Piumatti et al., 2018).

6. In Fig.8a, the YFP signal in the ventral region between the pons (2) and the striatum (5) as well in the region ventral to the hippocampus (4) is present in Galc-iKO mice TM P3->P35 but appears absent/reduced in mice ablated at P8. Is this due to the non-identical sagittal plane of the representative tissue slices? (details on the YFP quantification method are needed). Is it worth considering (and testing) the possibility that GALC deficiency affects differently neuronal populations in different CNS areas at different developmental times?

Detailed YFP quantification method has been added in the methods section. All the conditions for YFP signals were imaged with identical illumination, laser power and gain parameters using Lecia SP5 laser-scanning confocal microscopic analysis. The images were thresholded equally to achieve a binary image, subtracted by the background values obtained from the sections of non-YFP genotype, and then normalized to average of the control condition prior to statistical testing. The stitched whole brain images were acquired using z-stacks that were recorded utilizing sequential confocal images that were collected at 1 μm intervals covering 25 μm depth. As the reviewer pinpointed, it could be possible that the original representative images are not perfectly identical planes. Therefore, in this revised manuscript we have added serial section-images covering whole Galc-iKO brains in Fig. S4. As the reviewer commented, our data can't exclude a possibility that other brain regions could be affected by GALC deficiency at different times significantly. In the study, we have focused on the anatomic region with the most robust changes during our empirically defined critical period, P4-6.

Overall, the results presented here support the role of GALC to ensure proper development/morphology/survival of brainstem neurons in a cell-autonomous manner; however, they do not completely exclude a similar role in neurons of other brain regions (more subtle differences – dysfunction vs. cell death- may have been missed in this analysis). In particular, a comparison with more caudal CNS regions (i.e. spinal cord) that are normally highly and early affected in KD and express the highest psychosine load, would be needed to support the main claim of the paper.

We agree that there is a possibility of a similar role of neuron autonomous GALC in other CNS regions. As we responded above, we have focused on the brainstem region because that region showed the robust changes during the critical period P4-6 in the brain. As the reviewer suggested, in this revised manuscript we compared the overall psychosine level in multiple anatomic CNS regions including cervical spinal cord in Galc-iKOs. Unfortunately, we didn't have a chance to analyze psychosine in more caudal parts of spinal cord. Interestingly, the cervical spinal cord had similar psychosine levels compared to the brainstem, regardless of induction timing (Fig. 5f). As we also described in comment #2, we further found that the early Galc deletion results in more impact on the psychosine accumulation in the brainstem and cervical spinal cord than other brain regions, indicating a possibility that psychosine could affect spinal cord to a similar extent of brainstem. However, the analysis of psychosine in the neuron specific Galc-CKO induced at P3 did not show any distinct accumulation of psychosine in any CNS regions including brainstem and cervical spinal cord compared to WT (Fig. S8e), suggesting psychosine may not be generated by perinatal neurons, and neuronal GALC could have other function not related to psychosine accumulation. In neurons, lysosomes are positioned throughout the cell including proximal to the axons and dendrites (Parton, Simons, & Dotti, 1992). In these locations the lysosomes maintain homeostasis of the neuron and have been found to be recruited to dendrites during neuronal activity (Goo et al., 2017). Although the impact of lysosomes on dendritic function has yet to be fully elucidated, one possibility is that lysosomal dysfunction could affect dendritic remodeling due to the shift in the lipidome of the neuron (Olsen & Faergeman, 2017). Therefore, we hypothesize that Galc-deficient lysosome could affect primarily the lipidome of the neuron and defunct neuronal lysosomes. We will further study about the detailed role of neuron-autonomous GALC.

Also, the link between the neuronal GALC deficiency and psychosine in this model is still unclear. In this perspective, the comparative assessment of GALC activity and psychosine in selected CNS regions as well as the measurement of psychosine in tdTomato-Thy1+ neurons isolated from brainstem, spinal cord and more rostral regions would help clarifying this link

Unfortunately, it was impossible for us to generate new mice to isolate neurons in culture due to a 3 months complete laboratory shut down for Covid-19 and current phased and slow reopening. However, we would argue that these experiments, even if useful, would not necessarily be conclusive for two reasons. First, as mentioned above, psychosine was not differentially accumulated in P60 cerebral cortex, cerebellum, brainstem, and spinal cord of Thy1-Cre/ER^{T2} driven Galc-CKO induced at P3 (Fig. S8e). Second, we believe that psychosine is not only the toxin or the only pathogenetic mechanism in Krabbe pathogenesis. We recently proved that psychosine alone does not explain difference in some aspects of Krabbe disease severity (Weinstock et al.,

2020). We further show that, abnormal GalCer accumulation, the major substrate of GALC, causes cellular stress and a pro-inflammatory cell reaction (Weinstock et al., 2020). In addition, dysfunctional lysosome due to GALC deficiency could trigger secondary accumulation of other lysosomal substrates that may be detrimental to cell differentiation or survival as well.

These experiments led to the identification of a critical perinatal time window in which the developing murine brain is most vulnerable to GALC deficiency. As the author discuss, this period roughly corresponds (in terms of brain development) to the 2nd-3rd trimester of human gestation, further suggesting that in KD CNS pathogenesis starts before birth, possibly affecting neurogenesis and/or the function of early post-mitotic neurons and oligodendroglial progenitors. The brainstem is apparently most affected (even if the reason for this apparent selectivity is not elucidated) but other brain regions may be involved, although at lesser extent or at a different stage of CNS development.

Minor points

Discussion

“KD pathogenesis may begin very early in development, perhaps even prenatally”. There are indications of psychosine accumulation in human fetal brain tissues (Igisu, Science 1984)

We appreciate the reviewer’s correction and we have amended the previous report on the accumulation of psychosine in the fetal brain samples from Krabbe patients with the reference. Kobayashi et al (1988) found high level of psychosine accumulation in human fetal brain and spinal cord with KD, supporting the idea of prenatal disease development (Kobayashi et al., 1988). The study of Igisu and Suzuki (1984) specifically focused on the predominant accumulation of psychosine in the white matters of Galc-null mouse (twitcher), dog, and Krabbe patients, which has been added as the reference of “psychosine accumulates in white matter of the CNS, in a caudal-to-rostral pattern, parallel to the progression of myelin development (Igisu & Suzuki, 1984; Potter et al., 2013; Ricca et al., 2015)” in the result section.

“This finding may correlate with recent gene therapy trials for a number of LSDs in which increased enzyme activity was detected at the whole parenchymal brain level, but unfortunately resulted in minimal clinical improvements (ref 43, 44). This suggests that providing GALC in specific developmental period is critical to treat KD”. Providing sufficient levels of functional enzyme is as critical as the timing at which GALC is provided. Results of this study further suggest that >20% of normal levels (as measured in total brain lysates) are likely needed to prevent accumulation/reduce storage of

psychosine in brain tissues, even if region-specific differences have to be considered. This should be better investigated.

We agree that the return of GALC activity in moribund animals (from both early and late induced iKO mice) is indeed surprising. We also agree that a possible explanation for this phenomenon is that the returned GALC activity may not reach sufficient levels to achieve a clinical or phenotypic rescue. We have now presented this argument explicitly in the text as: providing sufficient GALC to specific cell types at an appropriate temporal period is critical to treat KD. However, regarding further studies exploring investigation of this phenomenon, we respectfully argue that this point is outside the scope of this study. The primary reasoning of pursuing this experiment in the first place was to try and explain the altered survival timelines of the two cohorts of early vs late induced Galc-iKO mice. As described, our findings show that both cohorts express similar levels of returned GALC at terminal survival, thus validating our employed technique and justifying the characterization of our altered survival further. To adequately address the issue of understanding the relationship of GALC expression to survival (i.e. what level of GALC expression is required, in what cells is it required, at what time is it then required in these cells), we anticipate many more studies would be required including BMT/gene therapy models. We plan to pursue these questions, in detail, in future studies. Furthermore, a recent publication of ours documents poor cross-correction of GALC in vivo (Weinstock et al., 2020). We have included this citation in the text.

“We show that brainstem development is dependent on intrinsic GALC expression by neurons. Specifically, brainstem neurons lacking GALC had perturbed differentiation and were in a more immature state”. This statement is not fully supported by the results presented here (see point 5-6)

We agree and have added more results showing 1) neuronal GALC expression is developmentally regulated in the brainstem (Fig. 6d-h), 2) Additional neuronal process marker, Tuj1 is reduced in the brainstem of Galc-KO (Fig. S3a). 3) Meticulous analysis of YFP neurons/axons in serial brain sections reveals that brainstem is the most affected during the P4-6 (Fig. S4). We will further study if immature neurons from the brainstem of Thy1-Cre/ER^{T2} driven Galc-CKO is more vulnerable to cellular stress as the disease progression.

Methods and figure legends

- Please indicate the statistical test used, the sample size, and the significance of the statistical comparisons for each graph/set of data in all the figure legends.

We have added the statistical method used, sample size, and significance in all figure legends as suggested.

- Cell counts and z-stacks (methods): please add details (see point 3)

We have added the detailed cell count and z-stacks in the methods section.

•Add method for YFP quantification (see point 6)

We have added YFP quantification method in the methods section.

References

- Allewelt, H., Taskindoust, M., Troy, J., Page, K., Wood, S., Parikh, S., . . . Kurtzberg, J. (2018). Long-Term Functional Outcomes after Hematopoietic Stem Cell Transplant for Early Infantile Krabbe Disease. *Biol Blood Marrow Transplant*, *24*(11), 2233-2238.
- Allewelt, H. B., Page, K., Taskindoust, M., Troy, J. D., Wood, S., Parikh, S., . . . Kurtzberg, J. (2016). Long-Term Functional Outcomes following Hematopoietic Stem Cell Transplantation for Krabbe Disease. *Biol Blood Marrow Transplant, Supplement*, S102-S103.
- Berg, S., Kutra, D., Kroeger, T., Straehle, C. N., Kausler, B. X., Haubold, C., . . . Kreshuk, A. (2019). Ilastik: Interactive Machine Learning for (Bio)image Analysis *Nat Methods*, *16*(12), 1226-1232.
- Bulfone, A., Smiga, S. M., Shimamura, K., Peterson, A., Puelles, L., & Rubenstein, J. L. R. (1995). T-Brain-1: A Homolog of Brachyury Whose Expression Defines Molecularly Distinct Domains within the Cerebral Cortex. *Neuron*, *15*(63-78).
- Castelvetri, L. C., Givogri, M. I., Zhu, H., Smith, B., Lopez-Rosas, A., Qiu, X., . . . Bongarzone, E. R. (2011). Axonopathy is a compounding factor in the pathogenesis of Krabbe disease. *Acta Neuropathol.*, *122*(1), 35-48.
- Englund, C., Fink, A., Lau, C., Pham, D., Daza, R. A. M., Bulfone, A., . . . Hevner, R. F. (2005). Pax6, Tbr2, and Tbr1 Are Expressed Sequentially by Radial Glia, Intermediate Progenitor Cells, and Postmitotic Neurons in Developing Neocortex. *J. Neurosci.*, *25*(1), 247-251.
- Fink, A. J., Englund, C., Daza, R. A. M., Pham, D., Lau, C., Nivison, M., . . . Hevner, R. F. (2006). Development of the Deep Cerebellar Nuclei: Transcription Factors and Cell Migration from the Rhombic Lip. *J. Neurosci.*, *26*(11), 3066-3076.
- Goo, M. S., Sancho, L., Slepak, N., Boassa, D., Deerinck, T. J., Ellisman, M. H., . . . Patrick, G. N. (2017). Activity-dependent trafficking of lysosomes in dendrites and dendritic spines. *J Cell Biol*, *216*(8), 2499-2513. doi:10.1083/jcb.201704068
- Hawkins-Salsbury, J. A., Shea, L., Jiang, X., Hunter, D. A., Guzman, A. M., Reddy, A. S., . . . Sands, M. S. (2015). Mechanism-Based Combination Treatment Dramatically Increases Therapeutic Efficacy in Murine Globoid Cell Leukodystrophy. *J. Neurosci.*, *35*(16), 6495-6505.
- Hevner, R. F., Shi, L., Justice, N., Hsueh, Y.-P., Sheng, M., Smiga, S., . . . Rubenstein, J. L. R. (2001). Tbr1 Regulates Differentiation of the Preplate and Layer 6. *Neuron*, *29*, 353-366.
- Hoshino, M. (2012). Neuronal subtype specification in the cerebellum and dorsal hindbrain. *Develop. Growth Differ.*, *54*, 317-326.
- Igisu, H., & Suzuki, K. (1984). Progressive accumulation of toxic metabolite in a genetic leukodystrophy *Science*, *224*(4650), 753-755.
- Kobayashi, T., Goto, I., Yamanaka, T., Suzuki, Y., Nakano, T., & Suzuki, K. (1988). Infantile and Fetal Globoid Cell Leukodystrophy: Analysis of Galactosylceramide and Galactosylsphingosine. *Ann. Neurol.*, *24*(4), 517-522.
- Marshall, M. S., Issa, Y., Jakubauskas, B., Stoskute, M., Elackattu, V., Marshall, J. N., . . . Bongarzone, E. R. (2018). Long-Term Improvement of Neurological Signs and Metabolic Dysfunction in a Mouse Model of Krabbe's Disease after Global Gene Therapy. *Mol. Ther., In Press*.
- Mullen, R. J., Buck, C. R., & Smith, A. M. (1992). NeuN, a neuronal specific nuclear protein in vertebrates. *Development*, *116*, 201-211.
- Olsen, A. S. B., & Faergeman, N. J. (2017). Sphingolipids: membrane microdomains in brain development, function and neurological diseases. *Open Biol*, *7*(5). doi:10.1098/rsob.170069
- Parton, R. G., Simons, K., & Dotti, C. G. (1992). Axonal and dendritic endocytic pathways in cultured neurons. *J Cell Biol*, *119*(1), 123-137. doi:10.1083/jcb.119.1.123

- Piumatti, M., Palazzo, O., Rosa, C. L., Crociara, P., Parolisi, R., Luzzati, F., . . . Bonfanti, L. (2018). Non-Newly Generated, "Immature" Neurons in the Sheep Brain Are Not Restricted to Cerebral Cortex. *J. Neurosci.*, *38*(4), 826-842.
- Potter, G. B., Santos, M., Davisson, M. T., Rowitch, D. H., Marks, D. L., Bongarzone, E. R., & Petryniak, M. A. (2013). Missense mutation in mouse GALC mimics human gene defect and offers new insights into Krabbe disease. *Hum. Mol. Genet.*, *22*(17), 3397-3414.
- Rafi, M. A., Rao, H. Z., Luzi, P., Curtis, M. T., & Wenger, D. A. (2012). Extended Normal Life After AAVrh10-mediated Gene Therapy in the Mouse Model of Krabbe Disease. *Mol. Ther.*, *20*(11), 2031-2042.
- Reddy, A. S., Kim, J. H., Hawkins-Salsbury, J. A., Macauley, S. L., Tracy, E. T., Vogler, C. A., . . . Sands, M. S. (2011). Bone Marrow Transplantation Augments the Effect of Brain- and Spinal Cord-Directed Adeno-Associated Virus 2/5 Gene Therapy by Altering Inflammation in the Murine Model of Globoid-Cell Leukodystrophy. *J. Neurosci.*, *31*(27), 9945-9957.
- Ricca, A., Rufo, N., Ungari, S., Morena, F., Martino, S., Kulik, W., . . . Gritti, A. (2015). Combined gene/cell therapies provide long-term and pervasive rescue of multiple pathological symptoms in a murine model of globoid cell leukodystrophy *Hum. Mol. Genet.*, *24*(12), 3372-3389.
- Suzuki, K. (2004). *Krabbe Disease: Myelin Biology and Disorders* (Vol. 2). USA: Elsevier.
- Weinstock, N. I., Shin, D., Dhimal, N., Hong, X., Irons, E. E., Silvestri, N. J., . . . Feltri, M. L. (2020). Macrophages Expressing GALC Improve Peripheral Krabbe Disease by a Mechanism Independent of Cross-Correction. *Neuron*, *S0896-6273*(20), 30238-30235.
- Weyer, A., & Schilling, K. (2003). Developmental and cell type-specific expression of the neuronal marker NeuN in the murine cerebellum. *J Neurosci Res*, *73*(3), 400-409. doi:10.1002/jnr.10655
- Wolf, N. I., Breur, M., Plug, B., Beerepoot, S., Westerveld, A. S. R., Rappard, D. F. v., . . . Bugiani, M. (2020). Metachromatic leukodystrophy and transplantation: remyelination, no cross-correction. *Ann Clin Transl Neurol.*, *7*(2), 169-180.
- Wright, M. D., Poe, M. D., DeRenzo, A., Haldal, S., & Escolar, M. L. (2017). Developmental outcomes of cord blood transplantation for Krabbe disease - A 15-year study. *Neurology*, *89*(13), 1365-1372.

Reviewers' Comments:

Reviewer #1:

Remarks to the Author:

The authors have addressed my concerns satisfactorily and I am happy for its publication in its current form

Reviewer #2:

Remarks to the Author:

The authors have comprehensively addressed all the comments, also including novel data and analysis that definitely improve the quality of the manuscript. This is particularly appreciated considering the difficult situation due to the COVID-19 lockdown.